# The malonyl/acetyl-transferase from murine fatty acid synthase is a promiscuous engineering tool for editing polyketide scaffolds
Lynn Buyachuihan [1,2], Simon Reiners[1,2], Yue Zhao[1] & Martin Grininger [1] ✉

Modular polyketide synthases (PKSs) play a vital role in the biosynthesis of complex natural products with pharmaceutically relevant properties. Their modular architecture makes them an attractive target for engineering to produce platform chemicals and drugs. In this study, we demonstrate that the promiscuous malonyl/acetyl-transferase domain (MAT) from murine fatty acid synthase serves as a highly versatile tool for the production of polyketide analogs. We evaluate the relevance of the MAT domain using three modular PKSs; the short trimodular venemycin synthase (VEMS), as well as modules of the PKSs deoxyerythronolide B synthase (DEBS) and pikromycin synthase (PIKS) responsible for the production of the antibiotic precursors erythromycin and pikromycin. To assess the performance of the MAT-swapped PKSs, we analyze the protein quality and run engineered polyketide syntheses in vitro. Our experiments include the chemoenzymatic synthesis of fluorinated macrolactones. Our study showcases MAT-based reprogramming of polyketide biosynthesis as a facile option for the regioselective editing of substituents decorating the polyketide scaffold.

Polyketides (PKs) are natural products with strong biological activities, such as the antibiotic erythromycin[1,2] or the immunosuppressant rapamycin[3,4]. They are produced by polyketide synthases (PKSs) through subsequent connection and modification of simple acyl building blocks[5,6]. Multienzyme type I PKSs can be divided into two types: (i) Modular PKSs consist of several modules that assemble into linear metastructures (Fig. 1A)[7,8]. They operate in a sequential manner with each module elongating the growing polyketide intermediate by two carbon atoms. Optionally, each module can further modify the intermediate before it is handed over to the next module. Essential domains for chain elongation include the acyltransferase (AT) domain, which selects the extender substrate (typically a malonyl- or methylmalonyl-CoA[9]), the ketoacyl synthase (KS) domain, which uses the extender substrate to elongate the PK intermediate, and the acyl carrier protein (ACP) shuttles the substrates and intermediates within and also between the modules[10]. If present, processing domains reduce and dehydrate the intermediate, which can introduce stereochemistry at the α- and β-position of the PK intermediate (Figs. 1A and S1). (ii) Iterative PKSs synthesize PKs via multiple cycles of chain elongation and processing is carried out by the same module[11–14].

Typically, modular PKSs exhibit a high product fidelity which can be attributed in part to the specificity of the substrate-selecting AT domains[15]. Different engineering attempts have been made, to alter the products of PKSs. Among these strategies, modification of the AT domains has demonstrated great potential, as it allows editing the polyketide scaffold with chemical entities that are introduced via the substrates[16]. Essentially, three strategies have been applied (Fig. 1B): (i) The AT domains have been engineered in the binding pocket to modulate substrate specificity[17–19]. (ii) The AT domains have been swapped with domains of different substrate specificity[20–24]. Recently, the substrate-specific AT domain of module 6 from DEBS was replaced by the promiscuous malonyl/acetyl-transferase (MAT) domain from the related murine fatty acid synthase (mFAS). This strategy allowed loading extender substrates that are not recognized by the native DEBS AT domain, such as fluorinated MalCoA analogs, resulting in macrolactone backbones similar to that of solithromycin[25,26]. (iii) Finally, the AT domains have been inactivated by point mutation and complemented with stand-alone trans-AT enzymes for loading the modular PKSs through protein–protein interactions, which among others, enabled biosynthesis of fluorinated erythromycin analogs[26–30].

[1]Institute of Organic Chemistry and Chemical Biology, Buchmann Institute for Molecular Life Sciences, Goethe University Frankfurt, 60438 Frankfurt am Main, Germany. [2]These authors contributed equally: Lynn Buyachuihan, Simon Reiners. ✉e-mail: grininger@chemie.uni-frankfurt.de

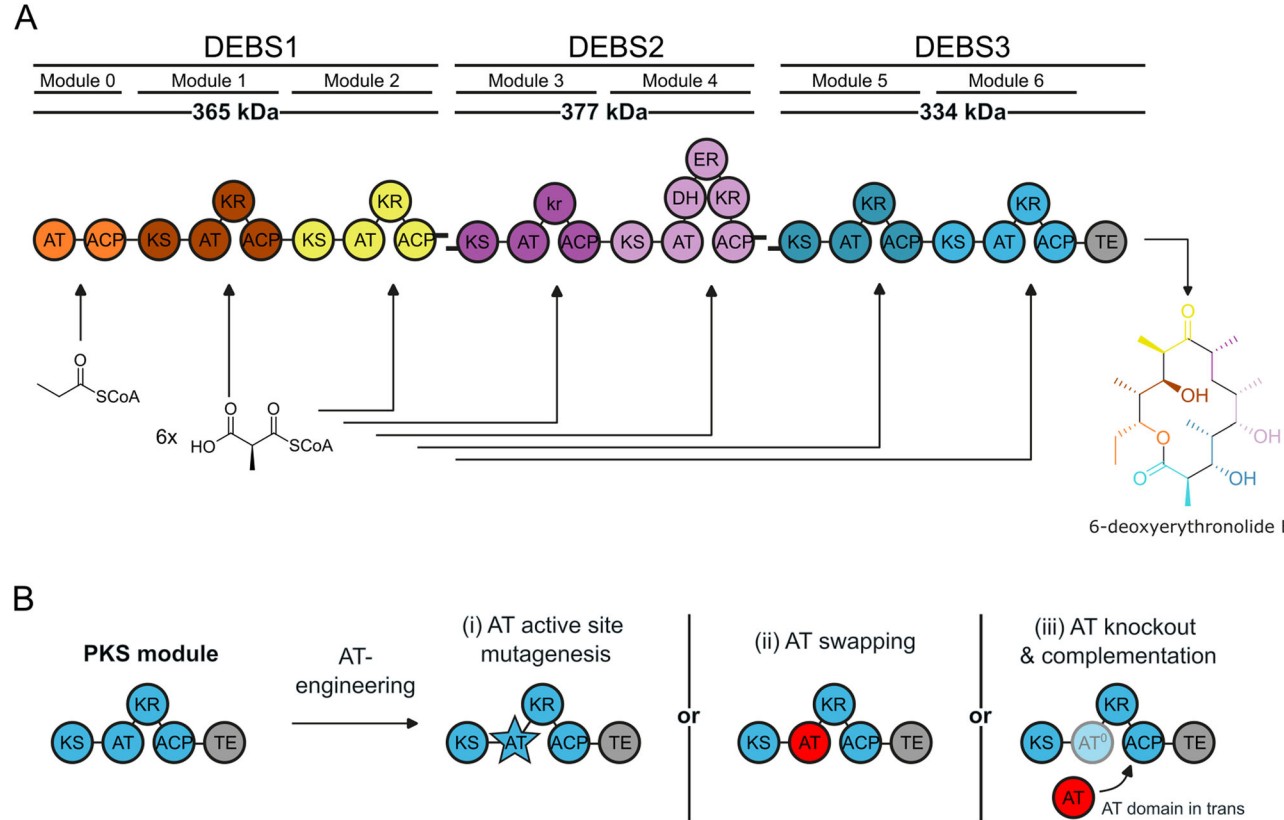

**Fig. 1 | Modular architecture of DEBS PKS and AT engineering approaches.**
**A** DEBS comprises three polypeptides DEBS1, DEBS2, and DEBS3, each harboring two elongation modules. DEBS1 possesses an N-terminal loading module (module 0) priming the PK synthesis with propionyl-CoA. The acyl intermediate is elongated six times with (2 S)-methylmalonyl-CoA. DEBS3 possesses a C-terminal TE domain responsible for the release of the product 6-deoxyerythronolide **B** The docking domains, represented by black tabs, allow for non-covalent interactions between the polypeptides. Domain annotation: KS ketosynthase, AT acyltransferase, KR ketoreductase, kr non-reductive ketoreductase-like domain, DH dehydratase, ER enoylreductase, ACP acyl carrier protein, and TE thioesterase. **B** AT-engineering approaches

illustrated on DEBS module 6. The extender substrate processed by the module can be altered by three strategies, namely (i) AT active site mutagenesis[18,19], (ii) AT swapping[25], and (iii) AT complementation in trans[27,28,30]. (i) AT active site mutagenesis involves the identification and mutation of residues which are responsible for AT substrate specificity. (ii) In AT-swapping approaches, the AT is exchanged with a foreign transferase, which is incorporated into the polypeptide (cis), replacing the native AT and exhibiting a different specificity profile. (iii) AT complementation involves inactivating the native AT and complementing it with another transferase as a standalone protein in trans.

The domain-swapping strategy benefits from transferring the functional properties of the external domain to the accepting PKS module, enabling the efficient regioselective editing of the substituents decorating the polyketide scaffold. However, the strategy is invasive, as the external domain introduces non-cognate domain-domain interfaces that can destabilize the PKS module. Furthermore, non-cognate transient transferase–ACP interactions arise during the loading of the intermodular ACP with the substrate[31]. The AT domains in modular PKSs are typically substrate selective and thus in control of the accurate selection of extender substrates supplied for the condensation reaction[32]. In iterative PKSs, AT domains are usually responsible for loading starter and extender substrates, which makes them inherently less specific. The MAT domain from mFAS is well characterized as a high substrate promiscuous and kinetically fast enzymatic domain, enabling efficient loading of various substrates, including non-, mono-, and disubstituted α-carboxyacyl-CoA extender substrates as well as acyl-CoA starter substrates with varying chain lengths and different stereochemistry and oxidation patterns at the α- and β-position[33,34]. Based on these properties and the high structural similarity of PKSs and animal FASs[35], we have recently achieved the chemoenzymatic synthesis of fluorinated macrolactones by AT/MAT-swapped FAS/PKS hybrid modules[25]. In the present work, we evaluate the scope of this approach to additional PKS modules and demonstrate that AT/MAT-swaps can be accomplished with high success rates. We engineer functional PKS modules and

assembly lines that generate new polyketides, including fluorinated derivatives of precursor macrolactones of the antibiotics methymycin and pikromycin. Specifically, we were working with the recently established split venemycin PKS (split VEMS) in vitro testbed[36], the engineered monomodular pikromycin synthase (PIKS) M5-TE[37], and the bimodular DEBS3 (Fig. 1A).

VEMS is a short modular PKS consisting of three modules (Fig. 2A)[38]. It utilizes the starter substrate 3,5-dihydroxybenzoic acid (DHBA) and elongates it twice using the extender substrate MalCoA, resulting in the production of venemycin **1**. In this study, we used the recently established split version of VEMS, where each module is located on an individual polypeptide chain (Fig. 2B)[36]. We aimed for the targeted modification of all three positions of the 2-pyrone product scaffold by introducing MAT swaps in all three modules of split VEMS (Fig. 2C). In this context, we tested different domain boundaries and found that the boundaries previously defined for DEBS module 6 (M6)[25], are most effective for VEMS as well.

Next, we used the validated boundaries to introduce AT/MAT-swaps to modules of the prominent PKSs PIKS (Fig. S2) and DEBS (Figs. 1A and S1A). Four hybrid constructs were targeted, systematically exchanging the specific AT domains of the engineered PIKS M5-TE and DEBS M5-M6-TE with the promiscuous MAT domain from mFAS (Fig. 2C). The functionality of the constructs was assessed with different extender substrates, yielding C12- and C14-membered macrolactone scaffolds with site-selective modifications.

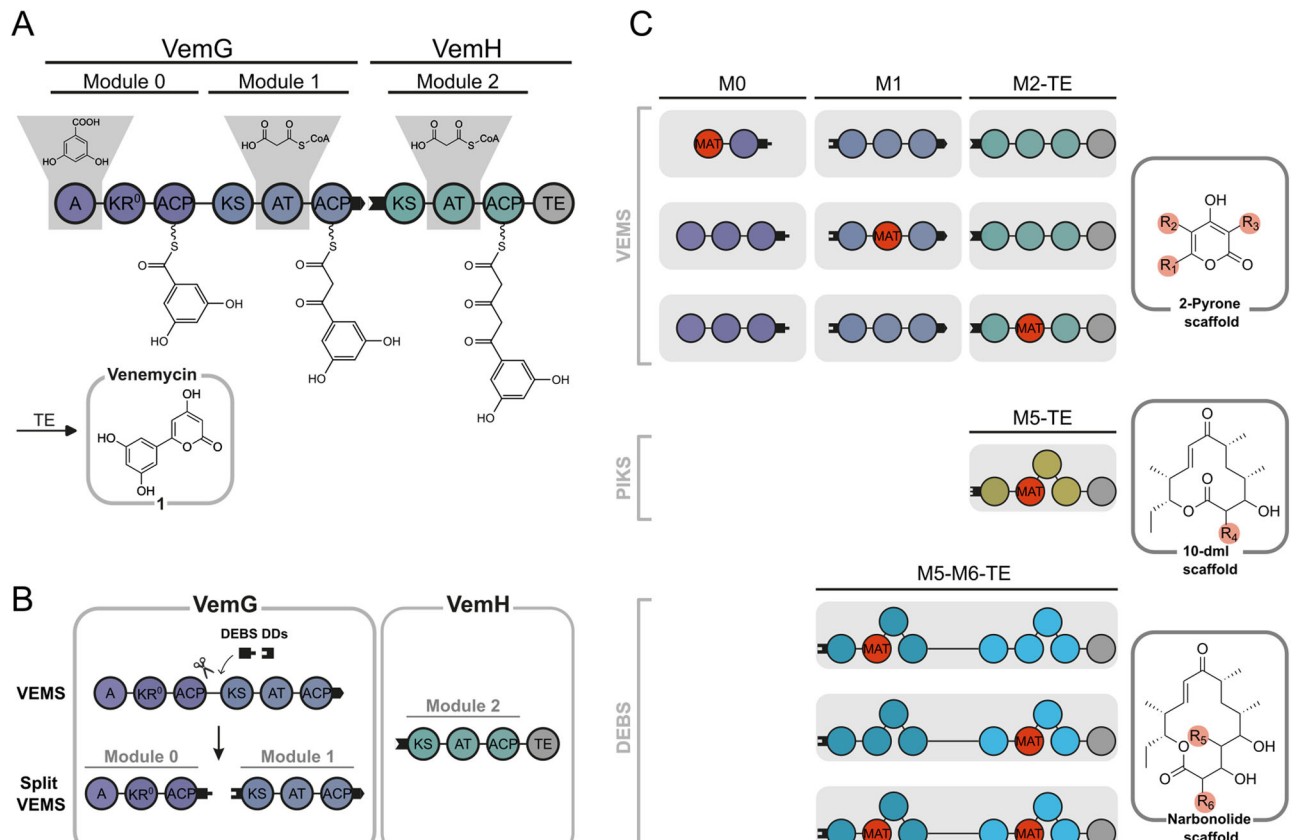

**Fig. 2 | Module and domain organization of VEMS, split VEMS, and AT/MAT-swapping in modules of VEMS, PIKS, and DEBS. A** VEMS is built by the two polypeptides VemG and VemH. The two elongation modules (module 1 and module 2), the loading module (module 0), the thioesterase domain, and the product venemycin **1** are illustrated. PK intermediates are attached to the respective acyl carrier protein (ACP). Black tabs depict docking domains. The starter substrate DHBA and the two extender substrates MalCoA are shown. Domain annotation: adenylation (A) domain; acyltransferase (AT) domain; ketoacyl synthase (KS) domain; acyl carrier protein (ACP); nonfunctional ketoacyl reductase (KR[0]) domain;

and thioesterase (TE) domain. **B** Module organization of the engineered PKS split VEMS[36] in which the polypeptide VemG was split into its loading module and the elongation module. Module 0 and module 1 were non-covalently connected via DEBS-derived docking domains. **C** AT/MAT-swaps were applied to all three modules of split VEMS, to an engineered form of PIKS module 5 with a C-terminal PIKS TE domain, and to the last polypeptide of the DEBS pathway DEBS3 (M5–M6-TE). Replacing the specific AT domains with the promiscuous mFAS-derived MAT domain allows the site-selective modification of the respective product scaffold marked in red.

## Results and discussion
### Engineering a PKS assembly line with a promiscuous loading module-
**Design of MAT-swapped loading modules for split VEMS.** First, we sought to introduce the polyspecific MAT domain into the VEMS loading module (M0), thereby creating a promiscuous hybrid loading module that allows for priming PK synthesis with various acyl-CoA starter substrates. The VEMS M0 comprises an adenylation (A) domain, responsible for the selection and activation of the starter substrate DHBA, which is subsequently loaded onto the ACP domain (Fig. 2A). Additionally, VEMS M0 features a nonfunctional KR domain[38] that is believed to play a structural role[39]. In the design of hybrid modules, it is inevitable to generate non-cognate domain-domain interactions (DDIs). Depending on the characteristics of the non-cognate interface, this can affect the performance of the engineered PKS. Due to the intricate nature of this issue, two different hybrid designs were evaluated (Fig. 3A): For one hybrid M0 (H1M0) the A-KR section was replaced by the MAT domain derived from mFAS and the VEMS ACP0 was further exchanged with the mFAS ACP, thereby ensuring native DDIs between MAT and ACP, both from mFAS, interacting during transacylation. In this case, a non-cognate interface arises during the translocation step between the mFAS ACP of M0 and the VEMS KS1 of the downstream M1 (Fig. S3). In the second hybrid M0 (H2M0), the VEMS ACP0 was retained providing

native interplay between the ACP0 and the KS1 during translocation. In this case, non-cognate DDIs occur during the transacylation between the mFAS MAT and the VEMS ACP0. The constructs were conceived based on a recent design of an mFAS-derived loading module from our lab. In this construct, the mFAS MAT domain was excised and fused to the mFAS ACP via a flexible 12-residue long linker[33]. Both hybrid modules H1M0 and H2M0 feature a C-terminal DEBS-derived docking domain to enable docking to the split VEMS M1, carrying the matching docking domain N-terminally.

**Evaluation of hybrid loading modules.** Both hybrid loading modules were produced in *E. coli*. While H1M0 was attempted to be purified via a single affinity tag, but did not achieve sufficient purity, H2M0 was received in high purity via tandem affinity chromatography, which was further validated via size exclusion chromatography (Fig. S4). To assess the functionality of the hybrid loading modules, they were tested within the context of a complete assembly line (HXM0–M1–M2-TE, Fig. 3A). It was assumed that the MAT domain loads its C-terminal ACP (mFAS ACP for H1M0 and VEMS ACP0 for H2M0) with a starter substrate, which then primes synthesis in split VEMS M1. Next, the starter substrate is elongated twice with MalCoA in M1 and M2-TE. The resulting triketide intermediate bound to the ACP2 of the final module M2-TE is then released under pyrone-ring formation. To prove the functionality of the

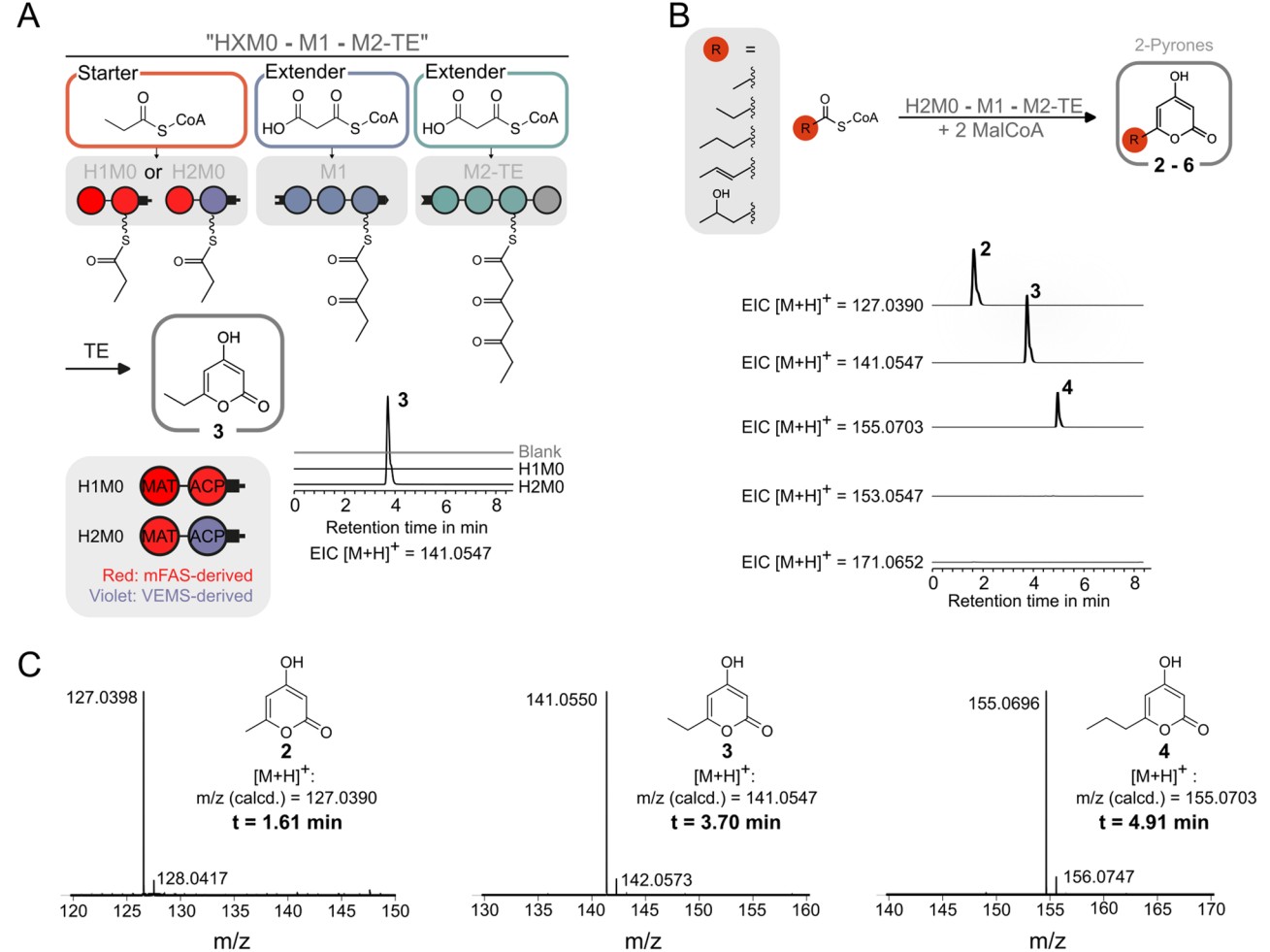

**Fig. 3 | Split VEMS with a promiscuous loading module. A** Two designs for a promiscuous loading module, H1M0, and H2M0 were evaluated. mFAS-derived domains are indicated in red. The production of 6-ethyl-4-hydroxy-2-pyrone **3** was analyzed via LC-HRMS (6-ethyl-4-hydroxy-2-pyrone **3** [M + H]⁺: m/z (calcd.) = 141.0547). Extracted ion chromatograms are shown for both PKSs using either H1M0 or H2M0. **B** The hybrid loading module H2M0 was challenged with different starter substrates, enabling the production of **2**, **3**, and **4**. Normalized extracted ion chromatograms (EICs) of 4-hydroxy-6-methyl-2-pyrone **2** (triacetic acid lactone,

TAL) and its derivatives are depicted (4-hydroxy-6-methyl-2-pyrone **2** [M + H]⁺: m/z (calcd.) = 127.0390, 6-ethyl-4-hydroxy-2-pyrone **3** [M + H]⁺: m/z (calcd.) = 141.0547, 4-hydroxy-6-propyl-2-pyrone **4** [M + H]⁺: m/z (calcd.) = 155.0703, (E)-4-hydroxy-6-propenyl-2-pyrone **5** [M + H]⁺: m/z (calcd.) = 153.0547, and 4-hydroxy-6-(2-hydroxypropyl)-2-pyrone **6** [M + H]⁺: m/z (calcd.) = 171.0652). **C** Mass spectra of compound **2** (found m/z = 127.0398), **3** (found m/z = 141.0550), and **4** (found m/z = 155.0696).

loading module, we decided to use propionyl-CoA (PrpCoA), which is well-accepted by the MAT domain[33]. Since the VEMS assembly lines may be primed by M1 AT-mediated loading of the extender substrate MalCoA and subsequent M1 KS-mediated decarboxylation, the use of acetyl-CoA (AcCoA) as a starter substrate could lead to ambiguous results. In contrast, successful incorporation of PrpCoA results in the formation of 6-ethyl-4-hydroxy-2-pyrone (**3**), and can be distinguished from decarboxylative priming with MalCoA leading to triacetic acid lactone (TAL, **2**). As analyzed by liquid chromatography-high resolution mass spectrometry (LC-HRMS), with PrpCoA as the priming substrate, expected compound **3** could be detected using H2M0 (Fig. 3A), preserving the native ACP-KS interface of chain translocation. We interpret the lack of activity in H1M0 containing PKS assembly lines as a result of the translocation reaction across the non-cognate interface. The condensation reaction (comprising chain translocation and elongation reaction) has been shown before by others and us to be rate-limiting in polyketide synthesis[40–45], such that any interference in the interaction between ACP and downstream KS can hinder substrate turnover. We link the lack of activity in H1M0-containing PKS assembly lines to the translocation reaction which involves the non-cognate mFAS ACP:VEMS KS1

interface. Studies have shown that the interference in the translocation reaction can throttle turnover[43–46], and, vice versa, that engineering strategies preserving the cognate interaction can maintain the efficiency of engineered PKSs[39,47,48].

**Exploiting the promiscuity of the MAT domain to produce 2-pyrone derivatives.** With a functional MAT-swapped loading module in hand, we sought to exploit the reported promiscuity of the MAT domain[33] to load the VEMS-based platform with a pool of starter substrates (Fig. 3B). A total of five starter substrates (acetyl-CoA, propionyl-CoA, butyryl-CoA, crotonyl-CoA, and 4-hydroxybutyryl-CoA), known to be accepted by the MAT domain[133], were tested. To investigate the tolerance of the downstream modules, these starter substrates were selected to possess varying chain lengths and oxidation patterns at the α and β positions. LC-HRMS analysis confirmed the production of 2-pyrone compounds (**2**–**4**, Fig. 3C) from three out of five tested starter substrates.

**MAT swaps in elongation modules of split VEMS**
Subsequently, we sought to replace the MalCoA-specific transferases (AT1 and AT2) of VEMS elongation modules (M1 and M2-TE) with the

promiscuous MAT domain. Previous studies have demonstrated that hybrid PKSs created by domain swapping frequently suffer from decreased catalytic activity and insolubility due to misfolding of the protein subunits[49,50]. Due to the lack of original structural data for VEMS, we utilized computational modeling (ColabFold[51]) to predict the structures of relevant VEMS subunits. The linker domain (LD) is of particular importance for AT/MAT-swaps, as it serves as a spacer between the KS and transferase (AT or MAT) domain. The LD is comprised of two distinct parts: $LD_1$, which is the part sandwiched between KS and transferase, and $LD_2$, which wraps back onto the KS surface and connects the transferase domain to the downstream ACP domain (Figs. 4A, S5, and S6). The essential question for the AT/MAT-swap is whether the transferase should be swapped, including or excluding the LD or whether it should be cut within the LD fold. In order to increase the chance of creating a functional hybrid construct, a broad range of domain boundaries for AT/MAT-swaps was tested. We categorized boundaries into three groups depending on how the LD was treated, i.e., whether it was retained from the acceptor module (VEMS M1 or VEMS M2-TE) or derived from the donor module (mFAS) (Figs. S5 and S6). In H1 constructs, the boundaries were positioned close to the core fold of the AT domain while preserving the LD ($LD_1$- and $LD_2$-part) of the accepting VEMS. Conversely, in H2 constructs, the LD was derived from the donating mFAS. Lastly, in the H3 group, the boundaries were set so that the $LD_1$-part originated from the acceptor module, while the $LD_2$-part was derived from the donor module (Fig. 4B, C).

### AT/MAT-swapped elongation module VEMS M2-TE.

For VEMS M2-TE, a total of 24 hybrid constructs M2*-TE were designed (Fig. S5), of which 6 constructs belong to the H1-group, 8 to the H2-group, and 10 to the H3-group, and subjected to test expression in *E. coli* (Tables S1–S4). Sodium dodecyl-sulfate polyacrylamide gel electrophoresis (SDS-PAGE) analysis revealed that certain constructs express, albeit at insufficient quantities (<0.1 mg/L *E. coli* culture) to allow protein isolation and analysis (Figs. S7 and S8). The difficulty in obtaining functional hybrid constructs in sufficient quantities underscores the intricate nature of the KS-AT didomain fold.

### AT/MAT-swapped elongation module VEMS M1.

Similarly, as for VEMS M2-TE, different boundaries for the AT/MAT-swap in the first elongation module M1 were tested. M1 engineering was performed in the context of the full-length polypeptide VemG (M0-M1) because we assumed the highest protein stability under native-like conditions. The boundaries of the best expressible hybrid construct should then be applied to the engineered M1 of split VEMS (M1). Four constructs were tested for expression in *E. coli* (2 of H1, 1 of H2, and 1 of H3 group, Figs. 4B and S6), of which one construct of the H1 group yielded enough protein for further analysis (Fig. S9). To confirm functionality, the AT/MAT-swapped M0–M1* was assembled with M2-TE to VEMS. LC-HRMS analysis confirmed the formation of venemycin **1** when using the substrates DHBA and MalCoA, indicating the functionality of the hybrid module (Fig. S10). These boundaries were then utilized to create a stand-alone AT/MAT-swapped M1* (Fig. S11) that is compatible with split VEMS. The AT/MAT-swapped hybrid module M1* was also active when used as a stand-alone module within the split VEMS testbed, as confirmed by-product formation (Fig. S10).

Since the polyspecific transferase offers promiscuous substrate loading, the hybrid assembly line was challenged with the non-native extender substrate MMalCoA, which can normally not be processed by split VEMS (Fig. 5). When using an extender substrate mixture of MalCoA and MMalCoA, the derivative 5-methyl-venemycin **7** was only produced by the assembly lines M0–M1* VEMS and M1* split VEMS, carrying the AT/MAT-swap, demonstrating the ability of the hybrid assembly line to incorporate non-native extender substrates. Of note, providing MalCoA and MMalCoA as extender substrates, M1* split VEMS also produced venemycin **1**, as a result of loading both malonyl

and methylmalonyl by the promiscuous MAT. The derivative 3,5-dimethyl-venemycin **8**, which would result from the incorporation of MMalCoA by both elongation modules, could not be observed due to the substrate-specific AT of M2-TE.

### Promiscuous assembly line with substrate-dependent starting points.

The mFAS-derived MAT domain possesses a dual functionality, meaning that it can prime but also continue fatty acid synthesis by loading the appropriate starter (AcCoA) or extender substrate (MalCoA). This property is not found in modular PKSs, in which transferases are more diversified and either load the starter for priming synthesis or the extender substrate for growing the polyketide intermediate. In this regard, the AT/MAT-swapped M1* offers the chance to harness M1* as an alternative starting position for polyketide synthesis. As a first test, we supplied propionyl-CoA to the hybrid M1* assembly line, which cannot be processed by split VEMS but is accepted by the MAT domain[33]. Since AcCoA may arise in situ during synthesis by decarboxylation of MalCoA, we chose to work with propionyl-CoA. Confirmed by LC-HRMS, the hybrid split VEMS assembly line produced the respective 2-pyrone **3** (Fig. 6A). Based on these data, we conclude that the production involves priming by M1*, followed by an elongation step by M1* before handing over the intermediate to M2-TE. We propose that the propionyl-loaded ACP transfers propionyl back to the intramodular KS domain, instead of translocating it to the downstream module, which is a mechanism observed in some modular PKSs in the context of "module stuttering"[52]. Subsequently, M2-TE catalyzes a second elongation step and the release of compound **3**.

The dual functionality of the MAT domain turns split VEMS into an assembly line with two priming points that can be utilized depending on the chemical structure of the starter substrate: DHBA-derivatives are activated by M0 (ref. 39) and forwarded to M1, while acyl-CoAs can be incorporated by MAT of M1*. Furthermore, the promiscuity of MAT in M1* enables the modulation of position 5 of the 2-pyrone scaffold when non-native extender substrates are used (Fig. 6B). We demonstrated the versatility of this assembly line by combining 5 different starter (DHBA, 3-hydroxybenzoic acid, acetyl-CoA, propionyl-CoA, and butyryl-CoA) and two extender substrates (MalCoA and MMalCoA) leading to 10 products with a 2-pyrone scaffold generated by one engineered PKS.

### AT/MAT-swaps in mono-modular PIKS system

To further interrogate the utility of the AT/MAT-swap strategy, we engineered a late-stage elongation module of the well-characterized PIKS, which is responsible for the production of the precursors of the antibiotics methymycin and pikromycin[53,54]. We chose the recently engineered PIKS module 5 (PIKS M5) that is C-terminally elongated with the PIKS M6 TE domain giving an overall domain structure of KS-AT-KR-ACP-TE (termed PIKS M5-TE)[37]. Domain boundaries for the AT/MAT-swap were chosen in accordance with the H1 boundaries previously established by Rittner and Joppe et al.[25] (Fig. S14), which also performed best in the comprehensive hybrid design screening of VEMS. PIKS M5-TE and the AT/MAT-swapped PIKS M5*-TE were produced in *E. coli*, with PIKS M5*-TE yielding about 50% of the amount of the non-swapped PIKS M5-TE (Fig. S15).

First, the activities of PIKS M5-TE and PIKS M5*-TE were tested chemoenzymatically by the in vitro product formation assay established by Sherman and coworkers[55] (Fig. 7A). Activated PIKS pentaketide was employed as the elaborate starter substrate mimic and MMalCoA was added as the native extender substrate of PIKS M5. The NADPH consumption of KR5 was monitored fluorometrically to evaluate the reaction progress (Fig. 7C). As confirmed by LC-HRMS, the hybrid module PIKS M5*-TE was able to produce the methymycin precursor 10-deoxymethynolide (10-dml, **14**), although at a reduced speed of 0.18 min$^{-1}$ compared to PIKS M5-TE (1.6 min$^{-1}$). Next, we challenged the hybrid PIKS M5*-TE with the non-native extender substrates MalCoA and fluoromethylmalonyl-CoA (FMMalCoA). The use of di-substituted FMMalCoA was first described in the context of a FAS/PKS hybrid of DEBS M6[25]. FMMalCoA gives access

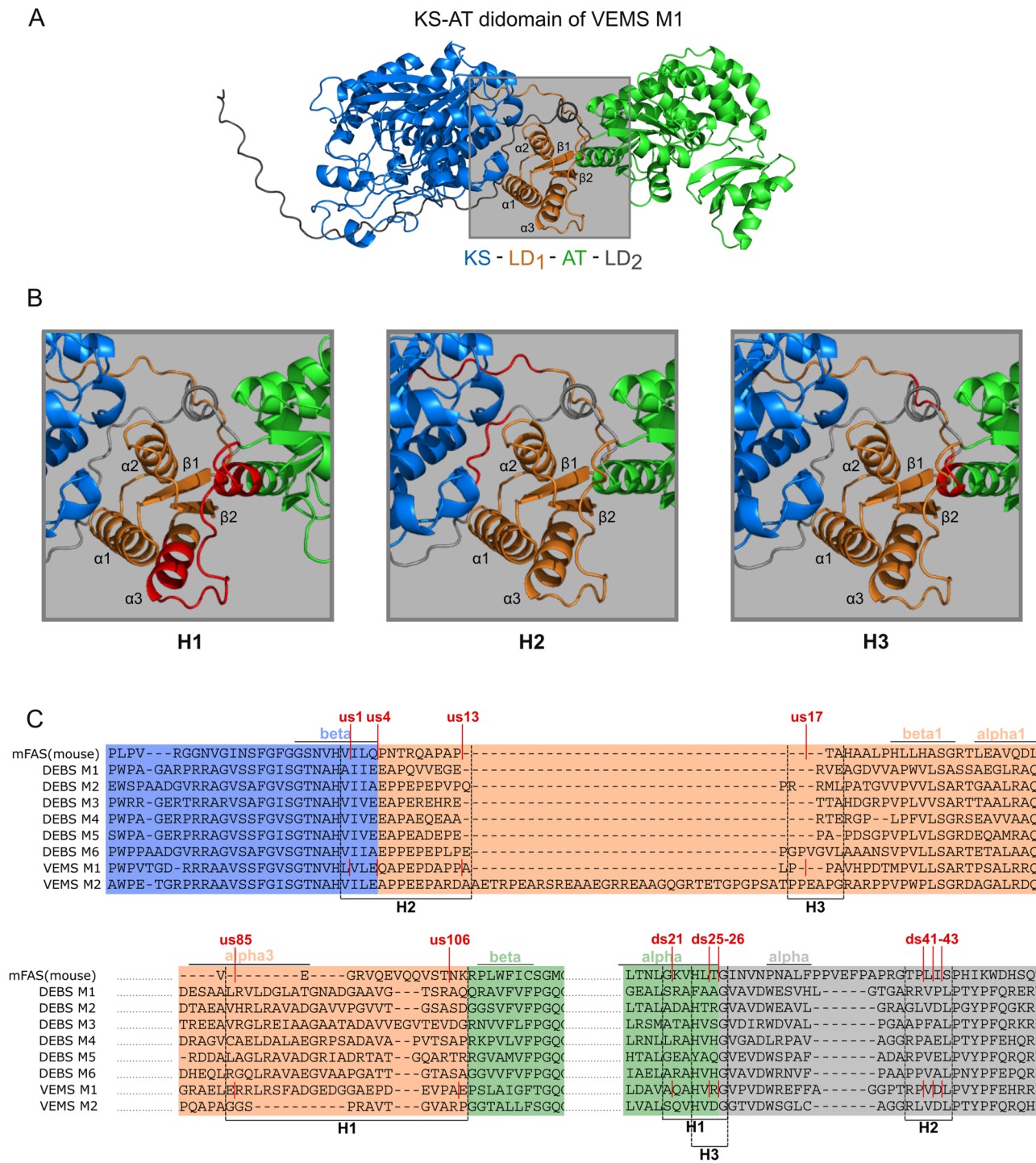

**Fig. 4 | Boundaries of AT/MAT-swapped VEMS constructs M0-M1\*. A** Structural model of the VEMS M1 KS-AT didomain fold (monomeric) predicted with ColabFold[51]. KS, LD$_1$, AT, and LD$_2$ domains are depicted in blue, orange, green, and gray cartoons, respectively. The LD is marked by a gray box. **B** Zoom into the LD. For a better orientation, LD$_1$-building secondary structure elements are numbered. MAT-swaps of the H1, H2, and H3 group are depicted. The residue ranges in which the junction sites were chosen differ for each hybrid group and are highlighted in red. **C** Shortened Multiple Sequence alignment (MSA) of the KS-AT didomain sequences

of mFAS and all elongation modules from DEBS and VEMS. The full MSA is provided in Fig. S6. Swap junctions upstream of the AT domain are termed usXX (XX is the residue position according to the first junction site us1, indicated in the MSA) and downstream of the AT domain dsXX in accordance with a previous study on AT swaps[49]. Secondary structure elements of the predicted structure of the VEMS M1 KS-AT didomain are indicated. The range in which the respective swap junctions (indicated in red) of the hybrid groups were chosen is framed with dashed lines.

to motifs found in solithromycin. Additionally, the incorporation of the F/Me-substitution demonstrates the ability of direct disubstitution of macrolactone scaffolds. The formation of the expected demethylated and fluorinated 10-dml derivatives 2-demethyl-10-dml **16** and 2-fluoro-10-dml

**17** was confirmed by LC-HRMS (see Fig. 7B for normalized EICs and S16 for mass spectra). NADPH monitoring revealed the highest turnover using the extender substrate MalCoA (0.48 min$^{-1}$), which is the preferred substrate of the MAT domain. Yet, 2-demethyl-3-oxo-10-dml **15** was identified as the

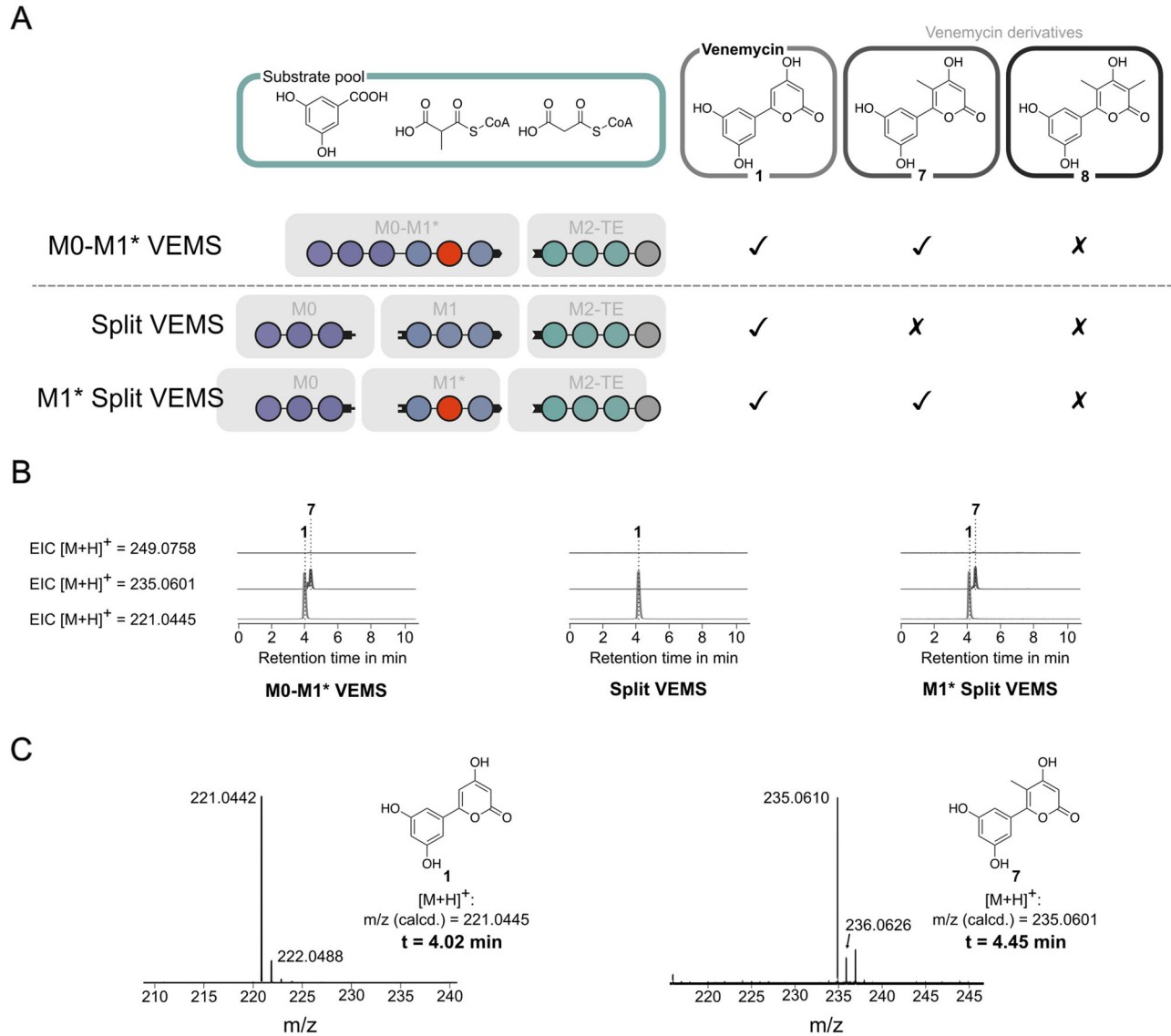

**Fig. 5 | M1* in VEMS and split VEMS. A** M1* can be operated in the context of VEMS and split VEMS. The resulting assembly lines are able to produce the native VEMS product venemycin **1**. Additionally, the MAT domain allows the incorporation of MMalCoA which is not accepted by the native AT1. This results in the production of a methylated venemycin derivative **7** which is not accessible by utilizing VEMS. **B** Normalized EICs of venemycin and its derivatives (venemycin **1** [M + H]⁺: m/z (calcd.) = 221.0445, 5-methyl-venemycin **7** [M + H]⁺: m/z (calcd.) = 235.0601, and 3,5-dimethyl-venemycin **8** [M + H]⁺: m/z (calcd.) = 249.0758. **C** Mass spectra of compound **1** (found m/z = 221.0442) and **7** (found m/z = 235.0610).

dominant product when incubated with MalCoA due to impaired acceptance of the demethylated intermediate by the KR5 domain such that the TE releases the non-reduced macrolactone **15**[56]. Unexpectedly, the formation of 2-fluoro-10-dml **17** was also observed with PIKS M5-TE, indicating that the native AT domain is "leaky" for the di-substituted FMMalCoA substrate.

## AT/MAT-swaps in bi-modular DEBS system

Next, we applied the AT/MAT-swap to DEBS3, the ultimate bimodular polypeptide of the 6-deoxyerythronolide B-producing DEBS assembly line. DEBS3 comprises two covalently connected elongation modules, module 5 (M5) and module 6 (M6), each possessing the KS-AT-KR-ACP module composition and a C-terminal TE domain facilitating product release. Overall, we designed three bimodular PKS/FAS hybrid proteins by systematically replacing the DEBS AT domains (AT5 and AT6) with the MAT domain using H1 boundaries (Fig. S14). In hybrid M5*–M6-TE, DEBS AT5 was replaced with the MAT domain, in hybrid M5–M6*-TE DEBS AT6 was replaced with the MAT domain, while in M5*–M6*-TE both AT domains were replaced with the MAT domain.

While hybrid M5*–M6-TE and M5–M6*-TE would allow the modification of position 4 or 2 of the C14-macrolactone product, respectively, hybrid M5*–M6*-TE simultaneously targets both positions. Recombinant production in *E. coli* yielded the hybrid M5–M6*-TE in high purity and similar yields to the WT DEBS3 (11 mg/L and 13 mg/L of purified protein, respectively, Fig. S17). In contrary, M5*-M6-TE was obtained in lower yield (0.4 mg per liter) compared to DEBS3, and the double hybrid M5*–M6*-TE could not be isolated at all.

As before, the activated PIKS pentaketide was employed as a starter substrate, and MMalCoA was added as the native extender substrate of DEBS AT5 and AT6 (Fig. 8A)[33]. The reduction of NADPH by KR5 and KR6 was monitored fluorometrically to evaluate the reaction progress (Fig. 8B). DEBS3 as well as both hybrids revealed to be active and produced the C14-macrolactone 3-hydroxy-narbonolide **18** by two elongations. Of note, both hybrids, as well as native DEBS3 used as a control, produced 10-dml **14** as a byproduct. Two pathways could account for the detected 12-membered macrolactone byproduct: Production of 10-dml involves either omission of M6 after elongation by M5, followed by direct release through the TE

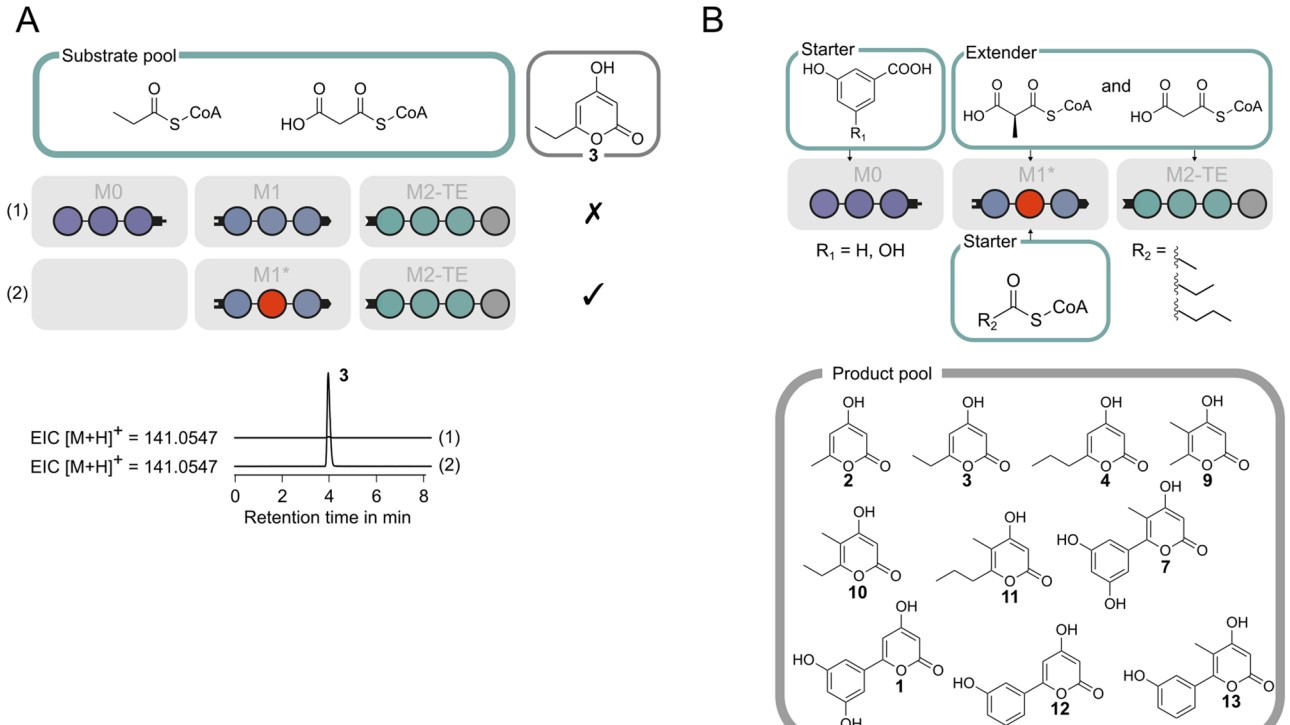

**Fig. 6 | Promiscuous assembly line with substrate-dependent starting points.**
**A** The dual functionality of the MAT domain can be used to start the PK synthesis at M1* and allows the utilization of propionyl-CoA as a starter substrate, resulting in the production of compound **3**, which would not be accessible via split VEMS. EICs

of 6-ethyl-4-hydroxy-2-pyrone **3** $[M + H]^+$: m/z (calcd.) = 141.0547. **B** Utilization of the MAT domain in the elongation module M1* allows starting the synthesis at two different positions of the assembly line, depending on the nature of the used starter substrate. EICs of the compounds are provided in Figs. S12 and S13.

domain of M6, resembling the natural PIKS pathway that accounts for 10-dml synthesis, or direct loading of KS6 of module 6 with the PIKS pentaketide followed by elongation and processing to 10-dml, a mechanism which was previously exploited by others and us[19,25,57].

Subsequently, the functional hybrid proteins were examined for their ability to produce derivatives of 14-membered macrolactone by providing 1:1 mixtures of extender substrates (MMalCoA/MalCoA or MMalCoA/FMMalCoA) and NADPH (Fig. 9A, B).

While for both substrate mixtures, native DEBS3 produced WT product 3-hydroxy-narbonolide **18** only (with 10-dml **14** observed as a byproduct via module skipping, see Figs. S18A and S19A, pathway (1)), the AT/MAT-swapped hybrid proteins led to more complex product outputs. M5*–M6-TE produced 4-demethyl-3-hydroxy-narbonolide **19** from the MMalCoA/MalCoA mixture (Fig. 9A) as well as 4-fluoro-3-hydroxy-narbonolide **20** from the MMalCoA/FMMalCoA mixture (Fig. 9B). In MMalCoA/MalCoA-containing samples, hardly any WT product **18** was observed in LC-HRMS and demethylated C-4 regioisomer **19** was the main product reflecting the efficiency of the MAT domain for MalCoA over MMalCoA (Fig. 9C for normalized EICs and S20 for mass spectra). In contrast, a mixture of WT product **18** and C-4-fluorinated product **20** was observed in samples containing MMalCoA/FMMalCoA substrate mixture (Fig. 9D for normalized EICs and S20 for mass spectra). The main product was compound **18**, which is in line with the diminished efficiency of MAT towards FMMalCoA observed before[25]. For MMalCoA/MalCoA substrate mixtures, we also identified 12-membered products 10-dml **14**, 2-demethyl-3-oxo-10-dml **15** and 2-demethyl-10-dml **16** from LC-HRMS, which was attributed to either skipping of hybrid module M5* and direct loading of the starter substrate to M6 and elongation with MMalCoA (see Fig. S18B, pathway (2)), or elongation with MalCoA in hybrid module M5*, optional KR5 skipping, followed by skipping of module M6 and direct TE-mediated offloading (see Fig. S18B, pathways (3) for **15** and (4) for **16**). We note that the chemical identity of compounds was determined by HPLC-HRMS, which does not give conclusive evidence of the regioselectivity of polyketide

modification. Given this limitation, NMR analysis of compounds will be necessary to confirm the modification at position 4 without ambiguity.

Similarly, hybrid M5–M6*-TE was subjected to mixtures containing MMalCoA/MalCoA or MMalCoA/FMMalCoA in order to prepare C-2 analogs 2-demethyl-3-hydroxy-narbonolide **21** and 2-fluoro-3-hydroxy-narbonolide **22** of WT product **18**. Mass peaks attributed to the respective C-2-demethylated or C-2-fluoromethylated compounds **21** and **22** were identified (Fig. 9C for **21** and Fig. 9D for **22**). Substantial shifts in retention time of approx. 0.5 min for 3-hydroxy-2-demethyl-narbonolide **21** and 0.3 min for 2-fluoro-3-hydroxy-narbonolide **22**, with respect to the C-4 compounds **19** and **20**, indicate that we produced different regioisomers. Chang and coworkers[30] have recently shown that C-2 and C-4 regioisomers of C14 macrolactones can be distinguished based on their retention time. However, again we note the NMR analysis will need to confirm suggested compound structures. Hybrid M5–M6*-TE produced the demethylated C-2 regioisomer **21** as the main product in mixtures containing MMalCoA/MalCoA. No WT product **18** was observed in LC-HRMS, but 2-demethyl-3-oxo-10 dml **15** and 2-demethyl-10-dml **16** were identified (see Fig. S18D) and explained by direct loading of the starter substrate onto KS6 of the hybrid module M6*, elongation with MalCoA and offloading of the non-reduced (KR-skipping) and the reduced compound (Fig. S18C, pathways (5) and (6)). The absence of 10-dml **14** in LC-HRMS indicated efficient translocation of the intermediate from M5 to M6*, loading of MalCoA in M6* only, and no M6*-skipping. A product mix of C14-membered products **18** and **22** (with 10-dml **14** as a byproduct, see Fig. S19) was obtained for MMalCoA/FMMalCoA assays similar to the product mix obtained by hybrid M5*–M6-TE.

## Conclusion
The implementation of the polyspecific MAT domain from mFAS in PKS engineering paves the way to the regioselective modification of PKS product by editing the substituents decorating the polyketide scaffold. In this study, we aimed to evaluate the broad applicability of AT/MAT-

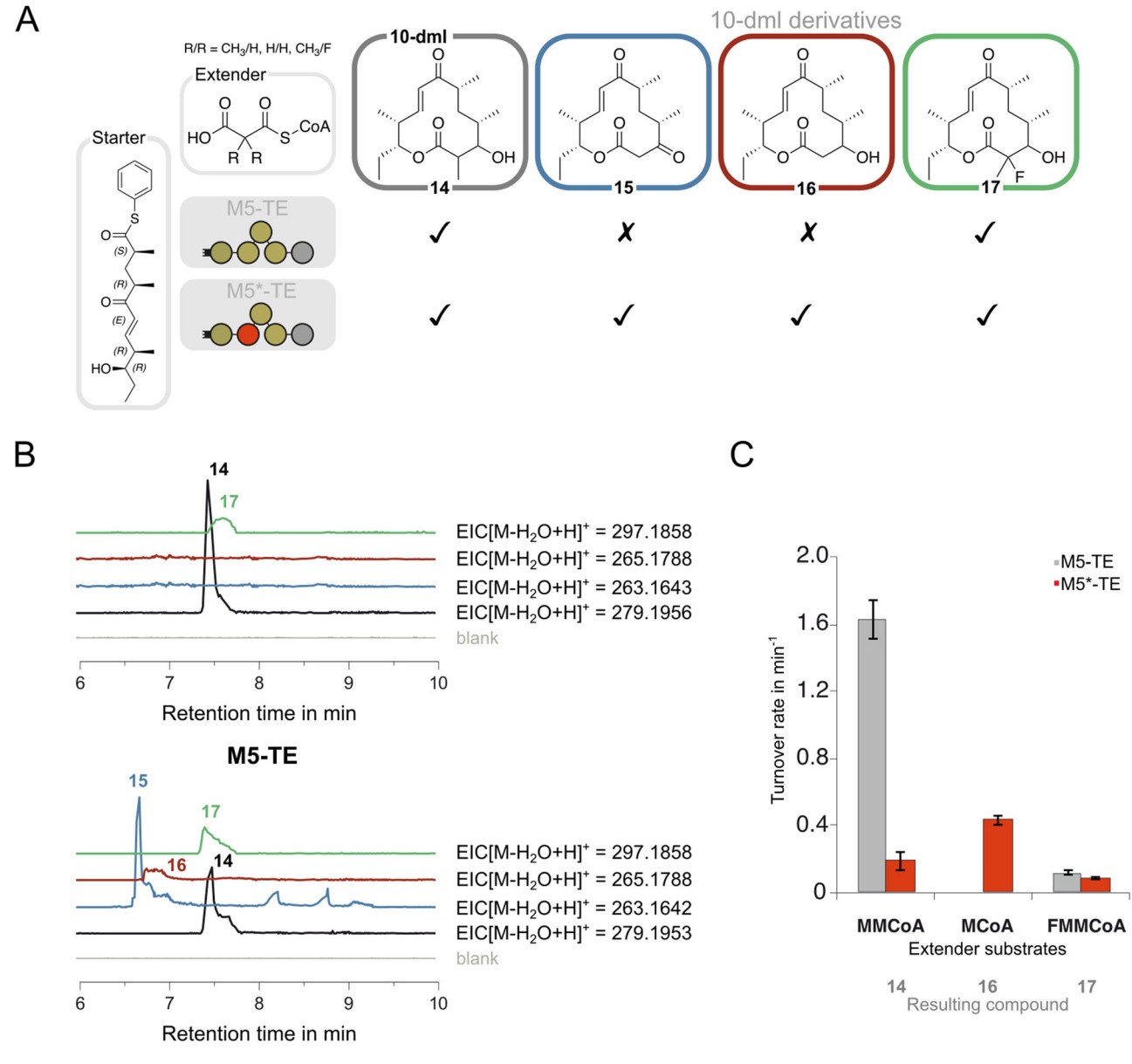

**Fig. 7 | Analysis of AT/MAT-swapped PIKS M5-TE in chemoenzymatic synthesis. A** Reaction scheme for the PIKS M5-TE-mediated elongation of PIKS pentaketide with native substrate MMalCoA or non-native substrates MalCoA and FMMalCoA to produce WT product **14** and derivatives thereof (**15–17**). **B** EICs of 10-dml and its derivatives detected by LC-HRMS (**14** [M + H-$H_2O$]$^+$: m/z (calcd.) = 279.1955, **15** [M + H-$H_2O$]$^+$: m/z (calcd.) = 263.1642, **16** [M + H-$H_2O$]$^+$: m/z (calcd.) = 265.1798, **17** [M + H-$H_2O$]$^+$: m/z (calcd.) = 297.1860). For both constructs, the peaks of **15–17** were normalized to 10-dml **14**. **C** Turnover rates measured by NADPH consumption of M5-TE- and M5*-TE-mediated reactions using different extender substrates.

swaps in PKS engineering. We focused on the in vitro analysis of engineered proteins, and considered product formation as a tool to determine the functionality of the hybrid constructs. Specifically, we have expanded the AT/MAT-swap to new PKS assembly lines (split VEMS and PIKS, DEBS used previously[25]) and interrogated the potential of the MAT's ability in the promiscuous loading of starter and extender substrates in PKS/FAS hybrid proteins.

Successful domain swaps could be designed at different locations within the assembly lines and in modules of different domain compositions. For the first time, loading modules (VEMS M0) and non-terminal modules (VEMS M1 and DEBS M5) have been addressed for the AT/MAT-swap to incorporate non-native starter and extender substrates during PK synthesis. Remarkably, the boundaries that exhibited the best performance in the

previously documented DEBS swap[25], consistently worked best across the tested boundaries in VEMS.

It should also be noted that this study demonstrates the ability of three modules of modular PKSs, PIKS M5, and DEBS M5 and M6 (DEBS M6 previously shown[25]) to elongate a growing polyketide chain with fluoromethylmalonyl. This suggests that, harnessing the MAT domain, the direct incorporation of disubstitutions into the polyketide scaffold appears to be a viable engineering strategy. Since most of the natural compounds are not directly administered as medicines but derivatized for use in therapy[58], derivatization is important to optimize a natural product's binding to its biological target, pharmacokinetic properties, and bioavailability. Based on our data, AT/MAT-swaps could constitute a viable engineering strategy for enabling not just mono-substitutions[59,60], but programmable and site-

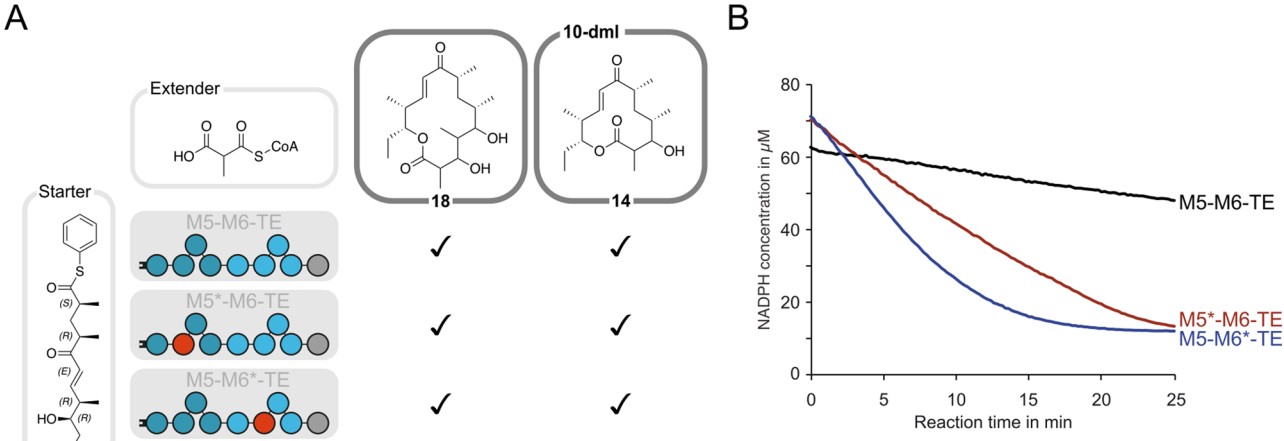

**Fig. 8 | Analysis of AT/MAT-swapped DEBS3-TE in chemoenzymatic synthesis.** Enzyme activity check. **A** Reaction scheme for the consecutive elongation of PIKS pentaketide with native substrate MMalCoA mediated by hybrids M5*–M6-TE and M5–M6*-TE to produce WT product **18**. 10-dml **14** is formed as a byproduct, which is explained by either skipping of M6 and direct translocation onto the TE domain or by direct loading of PIKS pentaketide onto the KS of M6. **B** Both hybrids demonstrate high turnover rates monitored by consumption of NADPH.

selective disubstitutions, such as fluoromethylations[61] shown in this study, or as a perspective gem-dimethylations[62]. We note that the acceptance of fluoromethylmalonyl by PIKS M5 (Fig. 7A) indicates that also ATs of PKSs may be suited to accept disubstituted extender substrates and could serve as an alternative to MAT. Previous data showed the general outstanding efficiency of mFAS MAT in the transacylation of non-canonical substrates[33], specifically, a nearly as high enzymatically efficiency for the non-cognate FMMalCoA ($k_{cat}$ 12.1 s$^{-1}$, $K_S$ 2.7 µM, $K_{ACP}$ 24.6 µM and $k_{cat}/K_S$ 4.5 × 10$^6$ M$^{-1}$s$^{-1}$)[25] than for AcCoA ($k_{cat}$ 99.2 s$^{-1}$, $K_S$ 12 µM, $K_{ACP}$ 265 µM, $k_{cat}/K_S$ 8.6 × 10$^6$ M$^{-1}$s$^{-1}$) and MalCoA ($k_{cat}$ 119 s$^{-1}$, $K_S$ 8.8 µM, $K_{ACP}$ 245 µM, $k_{cat}/K_S$ 14 × 10$^6$ M$^{-1}$s$^{-1}$)[33]. We have also determined kinetic parameters for PIKS M5 AT-mediated transacylation of the natural substrate MMalCoA to PIKS M5 ACP ($k_{cat}$ 0.75 s$^{-1}$, $K_S$ 96.2 µM, $K_{ACP}$ 330 µM, $k_{cat}/K_S$ 7.8 × 10$^4$ M$^{-1}$s$^{-1}$)[32], indicating that the enzymatic efficiency of mFAS MAT for the non-canonical substrate FMMCoA is 58 times higher than that of PIKS M5 AT for its natural substrate MMalCoA. Based on this comparison, we assume the good performance of PIKS M5, compared to the hybrid module, is not due to the enzymatic properties of the AT itself but to other factors, such as the maintained structural integrity of the module or the maintained cognate interaction between PIKS M5 AT and PIKS M5 ACP.

Based on our data, we can provide the following engineering guidelines for the AT/MAT-swap: (i) Prioritize the AT/MAT-exchange without flanking sequences (H1 boundaries), which in this study led to functional protein in five out of six modules, judged by yield and protein quality. Some PKS/FAS hybrid proteins with H2- and H3-designed boundaries were active in product formation assays (data not shown) but generally suffered from very low yields and poor purity. (ii). For none of the tested modules (VEMS M1 and M2-TE tested in this study, and DEBS M6-TE tested previously[25]), alternative domain boundaries have proven to be superior. Thus, AT/MAT-swaps using different boundaries are likely to offer low chances of success. (iii) Protein quality will suffer from the AT/MAT-swap, such that monitoring protein quality is important, especially when working with complex proteins and multiple AT/MAT-swaps per polypeptide. Intriguingly, even similar hybrid designs can lead to different outcomes, as demonstrated by the varying recombinant production levels of DEBS M5*–M6-TE (low protein yield) and DEBS M5–M6*-TE (high protein yield) (Fig. S17). We consider it a strength of in vitro studies to highlight such pitfalls, in this case, to illustrate that each protein design, regardless of how similar it is, will be subject to its own specific requirements. We note that an elaborate biosensor-guided method has recently been published, in which AT swaps can be quickly evaluated in hybrid protein quality. The approach enabled the screening of libraries of AT-exchanged hybrids in higher throughput, while finally identifying swap boundaries in agreement with our findings[49]. As a limitation of our study, we note that our engineering rules deviated from a dataset involving a total of six modules. The limited sample size derives from the goal of achieving a quantitative evaluation of protein performance in vitro.

With this approach, engineered PKSs can benefit from the broad substrate flexibility of the MAT domain which enables a more comprehensive expansion of the product spectrum compared to previous AT engineering efforts employing substrate-specific PKS AT domains[19,24]. Overall, our study demonstrates the broad potential of employing the substrate-tolerant MAT domain for the regioselective editing of polyketide scaffolds by side chain substitution during their biosynthesis. Of note, the accessible product scope can be limited by the tolerance of the downstream PKS domains to process the non-native intermediate. However, as a result of MAT's promiscuity, product mixtures are obtained when subjecting substrate mixtures to the hybrid PKSs. This should not necessarily be considered disadvantageous when working with a PKS assembly line with a single swap, as the product mixture would be just moderately complex and suitable for chromatographic separation. If demanded, the AT/MAT-swap approach can be further developed by utilizing specialized MAT variants, which were recently engineered through active site mutagenesis[33]. We would like to emphasize that broad substrate acceptance and high stability, both features observed for mFAS MAT, are generally regarded as good starting points for the engineering of proteins toward new functions[63]. The engineered M1* split VEMS proves to be a particularly interesting substrate-tolerant assembly line. With M0 of VEMS and the MAT in M1*, it contains two entry points to the assembly lines and possesses the ability to generate product pools in response to its substrate environment. Such designs may be developed into responsive elements in cell function and regulation.

In spite of the successes in AT/MAT-swaps, there are limits to the generation of FAS/PKS hybrids. We failed to obtain the hybrid modules VEMS M2*-TE and DEBS M5*–M6-TE, indicating that not all proteins can be harnessed for AT/MAT-swapping and that more than one swap per polypeptide lowers the chance to get a functional hybrid. Limitations may arise from the intertwined KS-LD-AT fold, and the general intricate structure of the PKS module.

## Methods
### Reagents
CloneAmp HiFi PCR Premix was from Takara. Primers were synthesized by Sigma Aldrich. For DNA purification, the GeneJET Plasmid Miniprep Kit, the GeneJET Gel Extraction Kit from ThermoFisher Scientific, and the

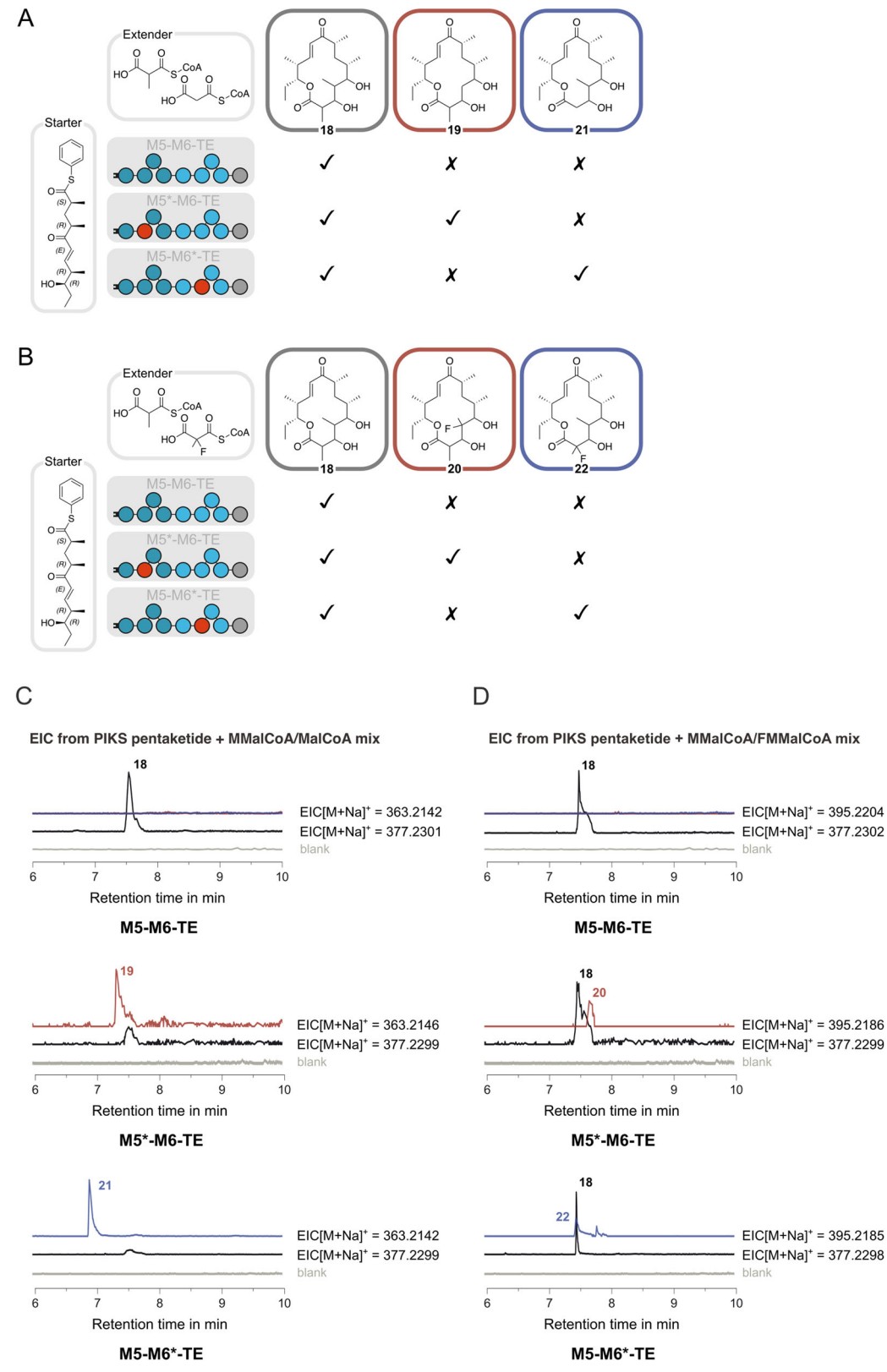

**Fig. 9 | Production of 14-membered macrolactones derivatized at C-2 or C-4.**
**A** Reaction scheme for the M5*–M6-TE- or M5–M6*-TE-mediated conversion of PIKS pentaketide with MMalCoA/MalCoA substrate mixture to produce macrolactone derivatives **19** and **21**. **B** Reaction scheme for the M5*–M6-TE- or M5–M6*-TE-mediated conversion of PIKS pentaketide with MMalCoA/FMMalCoA substrate mixture to produce macrolactone derivatives **20** and **22**. We note that the

chemical identity of compounds **19**–**22** was determined by HPLC-HRMS, which does not give conclusive evidence about the regioselectivity of polyketide modification. **C, D** EICs of macrolactones detected by LC-HRMS (**18** [M+Na]⁺: m/z (calcd.) = 337.2304; **19/21** [M+Na]⁺: m/z (calcd.) = 363.2142, **20/22** [M+Na]⁺: m/z (calcd.) = 395.2204). For each construct, peaks **19/21** and **20/22** are normalized to 3-hydroxy-narbonolide **18**, respectively.

Wizard® SV Gel and PCR Clean-up System from Promega Corp. were used. Cloning was performed with the In-Fusion HD Cloning Kit from Takara. NiCo21 (DE3) Competent Cells were from New England BioLabs. Stellar Competent Cells were from Takara, and One Shot BL21 (DE3) Cells were from ThermoFisher Scientific. All chemicals for buffer preparations were from Carl Roth GmbH ($KH_2PO_4$, $K_2HPO_4$, $NaH_2PO_4$, $Na_2HPO_4$, glycerol, disodium ethylenediaminetetraacetic acid (EDTA), VWR Chemicals (NaCl) and AlfaAesar (imidazole). Desthiobiotin was purchased from IBA Lifesciences, and biotin was from Abcr GmbH. For cell cultures, LB (Lennox) and 2× YT media were obtained from Carl Roth, Gibco™ Bacto™ Tryptone, and Gibco™ Bacto™ yeast extract was purchased from ThermoFisher Scientific. Isopropyl-β-D-1-thiogalactopyranoside (IPTG), ampicillin, and carbenicillin were from Carl Roth. Spectinomycin was from Sigma-Aldrich. His60 Ni Superflow Resin was from Takara, and Strep-Tactin Sepharose columns as well as Strep-Tactin®XT 4Flow® high capacity resin were from IBA Lifesciences. For anion exchange chromatography, the HiTrapQ HP column (column volume = 5 mL) was from Cytiva. Superose™ 6 Increase 10/300 GL column from Cytiva was used for SEC polishing and protein analysis. Proteins were concentrated with Amicon® Ultra centrifugal filters from Merck Millipore. Coenzyme A, 3,5-dihydroxybenzoic acid, 3-hydroxybenzoic acid, and adenosine-5'-triphosphate (ATP) were from Sigma-Aldrich. NADPH disodium salt was from F.Hoffmann-La Roche Ltd. Malonate, methylmalonate and magnesium chloride hexahydrate were from Carl Roth. Malonyl-CoA, methylmalonyl-CoA, and reducing agent tris-(2-carboxyethyl) phosphine (TCEP) was from ThermoFisher Scientific. Fluoromethylmalonyl-CoA was prepared as previously described[25]. Acetyl-, propionyl-, crotonyl-, 4-hydroxybutyryl- and butyryl-CoA were from Sigma Aldrich.

## Plasmids

AT/MAT-swapped constructs of VEMS PKS were generated via In-Fusion Cloning (Takara) from previously described plasmids. The DNA encoding PIKS proteins pikAIII (M5) and pikAIV (M6) (from *Streptomyces venezuelae* ATCC 15439 (DSMZ)), and DEBS protein DEBS3 (from *Saccharopolyspora erythraea* ATCC 11635 (DSMZ)) were amplified from genomic DNA by PCR and introduced into a pET22b(+) expression vector by In-Fusion Cloning (Takara). These expression plasmids (PIKS M5 (pAR328), PIKS M6-TE (pSR006), and DEBS3 M5-M6-TE (pAR268) as well as mFAS MAT (pAR246[33]) were used as a template to generate all engineered PIKS M5 and DEBS3 constructs of this study via In-Fusion Cloning (Takara). The resulting plasmids, cloning strategies, and primer sequences are further specified in Tables S1–S4. The plasmid sequences were verified by Sanger Sequencing (Microsynth Seqlab). All protein sequences of the constructs generated in this study are listed in Tables S5–S9. Construct designs for the boundary screening for AT/MAT-swaps in VemG M1 and VemH M2s are further illustrated in Figs. S5 and S6.

## Bacterial cell cultures

All PKS proteins were expressed and purified using similar protocols. H1M0, PIKS M5-TE- and VemG-based constructs were expressed in BAP1[64], VemH-based constructs and split VEMS M1-based constructs in NiCo21, and H2M0 and DEBS3-based constructs in BL21 cells. All proteins were expressed in the *holo*-form (to activate the ACP domain post-translationally with a phosphopantetheine arm). For expression in NiCo21 and BL21 cells, the construct-encoding plasmids were co-transformed with a plasmid encoding for the phosphopantetheine transferase Sfp from *Bacillus subtilis* (pAR357[33]). Cell cultures were grown on a 2 L scale in 2 × YT media (VEMS) or TB media (PIKS and DEBS) at 37 °C until an $OD_{600}$ of 0.3 was reached, whereupon the temperature was adjusted to 18 °C. At an $OD_{600}$ of 0.6, protein production was induced by adding 0.1 mM (VEMS) or 0.25 mM (PIKS and DEBS) IPTG, and the cells were grown for another 18 h at 140 rpm. Cells were harvested by centrifugation (5000 × *g*, 15 min), resuspended in lysis buffer (50 mM sodium phosphate, 10 mM imidazole, 450 mM NaCl, 10% glycerol, pH 7.6), lysed by French Press, and cell debris was removed by

centrifugation (50,000 × *g*, 45 min.) The cell lysate was subsequently purified using affinity chromatography. All constructs contained a C-terminal His-tag. VemG-based constructs, H2M0, and hybrid constructs PIKS M5*-TE and DEBS M5*-M6-TE contained an additional N-terminal Twin-Strep-tag for tandem affinity purification.

## Protein purification

For VEMS-based constructs, the purification procedure was as follows: For proteins containing just a C-terminal His-tag, the purification procedure was as follows: The supernatant was applied onto the column (5 mL Ni resin). A first wash step was performed with the above-mentioned lysis buffer (10 column volumes, CV), followed by a second wash step with 10 column volumes of wash buffer (50 mM sodium phosphate, 25 mM imidazole, 300 mM NaCl, 10% glycerol, pH 7.6). Proteins were eluted with 6 column volumes of elution buffer (50 mM sodium phosphate, 300 mM imidazole, 10% glycerol, pH 7.6). The eluate was purified by anion exchange chromatography using a HitrapQ column on an ÄKTA FPLC system (column volume 5 mL). Buffer A consisted of 50 mM sodium phosphate, 10% glycerol, pH 7.6, whereas buffer B contained 50 mM sodium phosphate, 500 mM NaCl, 10% glycerol, pH 7.6. Protein concentrations were determined with a Nanodrop. Samples were stored as aliquots at −80 °C until further use.

For proteins containing a C-terminal His- and an N-terminal Twin-Strep-tag, the purification procedure was as follows: The supernatant was applied onto the first affinity chromatography column (5 mL Ni resin). Washing was performed with 5 CV of lysis buffer (50 mM sodium phosphate, 10 mM imidazole, 450 mM NaCl, 10% glycerol, pH 7.6). Proteins were eluted with 2 × 2.5 column volumes of elution buffer (50 mM sodium phosphate, 300 mM imidazole, 10% glycerol, pH 7.6). The eluate was applied to the second affinity chromatography column (5 mL strep resin) and washed with 6 CV strep-wash buffer (50 mM sodium phosphate, 10% glycerol, pH 7.6). The target protein was eluted with 2 × 2.5 CV strep elution buffer (50 mM sodium phosphate, 2.5 mM desthiobiotin, 10% glycerol, pH 7.6). The eluate was purified by anion exchange chromatography using a HitrapQ column on an ÄKTA FPLC system (column volume 5 mL). Buffer A consisted of 50 mM sodium phosphate, 10% glycerol, pH 7.6, whereas buffer B contained 50 mM sodium phosphate, 500 mM NaCl, 10% glycerol, pH 7.6. Protein concentrations were determined with a Nanodrop. Samples were stored as aliquots at −80 °C until further use. VEMS-based constructs H2M0 and M1* (YZ046) were further polished and analyzed via size exclusion chromatography (SEC) using an ÄKTA FPLC system with a Superose 6 Increase 10/300 GL column from Cytiva (buffer: 50 mM sodium phosphate, 500 mM NaCl, 10% glycerol, pH 7.55). The SEC profiles are provided in Figs. S4 and S11.

For PIKS and DEBS proteins, the purification procedure was as follows: The cell lysate was applied onto the column (5 mL Ni resin), and the flowthrough was discarded. The proteins were washed with (i) 5 CV lysis buffer (25 mL), (ii) 2 CV wash buffer (10 mL; 50 mM sodium phosphate, 30 mM imidazole, 450 mM NaCl, 10% glycerol, pH 7.6) and the target protein was eluted with 2.5 CV elution buffer (12.5 mL; 50 mM sodium phosphate, 300 mM imidazole, 450 mM NaCl, 10% glycerol, pH 7.6). Proteins bearing a Twin-Strep-tag were additionally purified via Strep-Tactin XT 4Flow high-capacity resins (2 mL). The eluent was transferred to the Strep-Tactin XT resin, washed with 5 CV strep wash buffer (10 mL; 50 mM sodium phosphate, 10 mM imidazole, 450 mM NaCl, 10% glycerol, pH 7.6) and eluted with 2.5 CV strep elution buffer (5 mL; 50 mM sodium phosphate, 50 mM biotin, 10 mM imidazole, 450 mM NaCl, pH 7.6). Proteins were further polished and analyzed by SEC using a Superose 6 Increase 10/300 GL column equilibrated with SEC buffer (250 mM potassium phosphate, 10% glycerol, pH 7.0). Target protein fractions were pooled, concentrated to 10–20 mg mL$^{-1}$ using Amicon® Ultra centrifugal filters (100 kDa MWCO), flash frozen in liquid nitrogen and stored at −80 °C until further use.

The enzymes PrpE and MatB (starter and extender substrate regeneration system) were purified as described previously[65,66]. Protein

 

concentrations were determined with a Nanodrop. Samples were stored as aliquots at −80 °C until further use.

## Product formation assay

For VEMS-based assembly lines, reactions were carried out in a 40 µL scale. Reaction solutions were prepared using the reaction buffer (400 mM sodium phosphate, 10% glycerol, and pH 7.2). The reducing agent TCEP was used in a final concentration of 5 mM. In situ extender substrate generation was performed using 10 µM MatB, 1 mM CoA, and 10 mM malonate or 5 mM malonate and 5 mM methylmalonate. ATP and MgCl$_2$ were provided in a final concentration of 9 mM. The starter substrates DHBA and 3-hydroxybenzoic acid were used at a final concentration of 0.75 mM, while CoA-based starter substrates (acetyl-CoA, propionyl-CoA, crotonyl-CoA, 4-hydroxybutyryl-CoA, and butyryl-CoA) were used at a final concentration of 1 mM. The enzymes constituting the assembly line were used in a final concentration of 8 µM. The reaction mixture was incubated overnight at room temperature and extracted two times with 450 µL ethyl acetate. Dried samples were reconstituted in 100 µL methanol and analyzed by HPLC-HRMS in positive mode. Components were separated over a 16 min linear gradient of acetonitrile from 5% to 95% in water. m/z ratio of venemycin **1** $[M + H]^+$ m/z (calcd.) = 221.0445, 4-hydroxy-6-methyl-2-pyrone **2** $[M + H]^+$: m/z (calcd.) = 127.0390, 6-ethyl-4-hydroxy-2-pyrone **3** $[M + H]^+$: m/z (calcd.) = 141.0547, 4-hydroxy-6-propyl-2-pyrone **4** $[M + H]^+$: m/z (calcd.) = 155.0703, (E)-4-hydroxy-6-propenyl-2-pyrone **5** $[M + H]^+$: m/z (calcd.) = 153.0547, and 4-hydroxy-6-(2-hydroxypropyl)-2-pyrone **6** $[M + H]^+$: m/z (calcd.) = 171.0652, 5-methyl-venemycin **7** $[M + H]^+$: m/z (calcd.) = 235.0601, 3,5-dimethyl-venemycin **8** $[M + H]^+$: m/z (calcd.) = 249.0758. For EIC blot display, the m/z tolerance was set to 0.003.

For DEBS- and PIKS-based constructs all assays were performed in SEC buffer (250 mM potassium phosphate, 10% glycerol, pH 7.0) at 25 °C and a total volume of 20 µL. The assay was conducted in 384-well Small Volume HiBase Microplates (Greiner Bio-one) with ClarioStar microplate reader (BMG labtech). Settings: absorption: 348-20 nm, emission 476-20 nm, gain: 1500, focal height: 12.4 mm, flashes: 12, orbital averaging: off. The enzymes (PIKS M5-TE, PIKS M5*-TE, DEBS M5–M6-TE, DEBS M5*–M6-TE, and DEBS M5–M6*-TE) were prepared as a 4-fold stock solution (16 µM) (Sol 1), PIKS pentaketide was also prepared as a 4-fold stock solution (4 mM) (Sol2), and the X-CoA (400 µM)/NADPH (120 µM) mixture was prepared as a 2-fold stock solution (Sol 3). For testing bimodular DEBS3, Sol 3 comprised a 1:1-mixture of X-CoA substrates (MMalCoA/MalCoA and MMalCoA/FMMalCoA; total concentration: 400 µM) and NADPH (120 µM). 5 µL priming substrate (Sol 2) were mixed with 5 µL enzyme solution (Sol 1) and reaction was initiated by adding 10 µL of X-CoA/NADPH mix (Sol 3). Assay components were provided at final concentrations of 4 µM (enzyme), 1 mM (PIKS pentaketide), 200 µM XCoA, and 60 µM NADPH. Reaction progress was monitored fluorometrically for 30 min and converted into concentrations using a NADPH calibration curve. Afterward, reaction mixtures were immediately extracted with ethyl acetate (3 × 300 µL), the organic solvent was removed in a SpeedVac *in vacuo*, and dried samples were subsequently dissolved in 50 µL methanol and centrifuged at 20,000 × g for 20 min. Forty microliter supernatant was measured on HPLC-ESI-MS using the Ultimate 3000 LC (Dionex) system equipped with an Acquity UPLC BEH C18 (2.1 × 50 mm, particle size 1.7 µm, Waters) and connected to an AmaZonX (Bruker) or Impact II qTof (Bruker). m/z ratios of macrolactone compounds: **14** $[M + H-H_2O]^+$: m/z (calcd.) = 279.1955, **15** $[M + H-H_2O]^+$: m/z (calcd.) = 263.1642, **16** $[M + H-H_2O]^+$: m/z (calcd.) = 265.1798, **17** $[M + H-H_2O]^+$: m/z (calcd.) = 297.1860, **18** $[M+Na]^+$: m/z (calcd.) = 377.2304, **19/21** $[M+Na]^+$: m/z (calcd.) = 363.2142, and **20/22** $[M+Na]^+$: m/z (calcd.) = 395.2204. For the EIC blot display, the m/z tolerance was set to 0.003.

## Bioinformatical analysis

Structure predictions were performed with ColabFold[51] using default settings. The structures were predicted without template information. MSA options were set as follows: MMseqs2 (UniRef + Environmental) was chosen as MSA mode and unpaired + paired as pair mode, and the model type was set to auto. Amino acid sequences used for structure prediction are provided in Table S10. AlphaFold error estimates are provided in Figs. S21 and S22.

## Reporting summary

Further information on research design is available in the Nature Portfolio Reporting Summary linked to this article.

## Data availability

The authors declare that all data supporting the main findings of the article, including materials and methods, are described in the paper or supplementary information. Alternatively, the data are available from the corresponding author on request.

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

## Acknowledgements

Support for this work was received from the DFG grant GR3854/6-2 and 10-1. We thank H. Bode for use of their LC-HRMS instrument. We thank M. Joppe for starting the project with us (DEBS construct design) and Prof. D. Sherman for providing the substrate PIKS pentaketide.

## Author contributions

L.B., S.R., and M.G. wrote the paper. For VEMS-based constructs, L.B. conceived and supervised the project and designed the constructs. L.B. and Y.Z. cloned and purified the constructs. L.B. analyzed the data. For PIKS- and DEBS-based constructs, S.R. conceived the project, designed, cloned and purified the constructs, and analyzed the data. M.G. designed the research and analyzed the data.

## Funding

## Competing interests

The authors declare no competing interests.
