## [Peer Review File · Communications Chemistry]

Reviewers' comments:

Reviewer #1 (Remarks to the Author):

Grininger and coworkers report the implementation of a promiscuous MAT domain to replace the native AT machinery of the venemycin PKS machinery and the native AT machinery of both Pik and DEBS platforms. Beginning with a small PKS system, MAT is swapped into the VEMS assembly line with adjusted linker regions to observe non-native starter and extender unit integration for an array of diverse pyrone products. MAT swaps are then tested in terminal modules of Pik and DEBS assembly lines where integration of MalCoA and FMALCoA are observed. Their experiments show that MAT swaps result in promiscuous integration of starter and extender units, provided adequate care is taken to engineer the linking regions between PKS domains. The results in this manuscript are carefully prepared, and the arguments are thoughtfully laid out. This manuscript is therefore recommended for publication following the address of the following minor suggestions and concerns.

Line 61: On what basis do the authors claim that fluorination is the most important application of trans-AT machinery? There is no guarantee that structural similarity to solithromycin will result in medicinally useful compounds, especially since solithromycin was disqualified from clinical trials and nafithromycin has no fluorine. Please consider elaborating on this importance.

Line 78: A brief callout description of known substrates would be useful for the reader.

Line 133: How can H1M0 have been purified and yet “not be obtained in purity”?

Line 138: It was assumed that the MAT could load the substrate onto M0 ACP despite the non-cognate interaction. This might be a reach. Is the lack of activity with AcCoA observed by H1M0 due to the MAT-ACP0 disruption or the ACP0-KS1 disruption? Please consider elaborating on why this assumption is valid.

Line 182: Sorry if I missed this, but consider explaining where 6H1, 8H2 and 10H3 constructs came from.

Line 194: Consider explaining your cutoff for sufficient protein to purify/further analysis.

Figure 3B: The LCMS plots indicate that compounds 2, 3, and 6 were successfully produced, but the checks/ticks indicate the production of 2, 3, and 5. Likewise, the commentary text (page 5, line 150 and page 18, line 587) indicates 2, 3, and 4. Please clarify. Maintain the same order in the legends as in the LCMS chromatogram traces. Propionyl and 4-hydroxybutyryl-CoA are swapped in order.

Line 205: Consider adding a brief comment on the relative production of 5-methyl-venemycin compared to venemycin production.

Line 209: Perhaps not relevant for publication, but I'm curious if you tested a MAT swap in AT1 and AT2 simultaneously to see if you could produce 8?

Line 228: This "loading and extension" activity by M1* seems reminiscent of module stuttering, as mentioned by Keatinge-Clay 2012 and others. Consider commenting on this.

Line 258: Consider elaborating a little more on the relevance of fluoromethylmalonyl-CoA.

Line 264: This is unexpected indeed. Consider describing the implications of this somewhere in the manuscript. For example, does this suggest that the chimeragenesis approach might not always be necessary? Or that the Pik M5 AT could serve as an alternative to MAT?

Line 283: Consider explaining why swapping M5 would cause such a drop in purified protein.

Line 323: Shifts in retention time can not be conclusive evidence of regioselective differences. While differences certainly suggest that regiospecificity has changed, only NMR and X-Ray experiments conclusively prove this. Please use language that "suggests" the production of the supposed analogs. Different retention times are "consistent with" regioisomers.

Line 353: "led" instead of "let"

Line 369: "Broad flexibility of the MAT domain". Ok, but this needs context. It is only broader with respect to fluorinated malonyl-CoAs. There are likely many extenders MAT cannot use that native or engineered ATs can. How well would this approach work with assembly lines that contain malonyl-

CoA specific ATs that might compete with the FMM-CoA? Or indeed assembly lines that are as promiscuous as the Pik Module 5 AT?

Line 378: If MAT needs to be engineered to be more specific within a biosynthetic pathway, how is it superior to assembly line ATs, which are already extender unit specific?

Supplemental Figure 1B: why does the stereochemistry of the alpha carbon change after elongation and again after reduction? Shouldn't this center remain unchanged?

Supplemental Figure 3: The connection of substrates to the KS should occur on a straight line, as this bond is the direct carbon-sulfur bond of cysteine rather than a pantetheinylated sulfur bond of an ACP.

Supplemental Figure 14: it is optimistically described that the presence of 10 in a sample expecting to produce 4 "cannot be ruled out", but given that other experiments (Figure S14,C,right) present a peak with an identical retention time, it seems extremely likely that this peak is in fact 10. Coupled with the fact that this second peak could not be produced in the absence of mmCoA, this seems obvious. It would not be a loss to say so.

Reviewer #2 (Remarks to the Author):

Communications Chemistry (Submission ID COMMSCHEM-24-0049)

The malonyl/acetyl-transferase from murine fatty acid synthase is a promiscuous engineering tool for editing polyketide scaffolds

Buyachuihan L et al

The authors of this manuscript titled "The malonyl/acetyl-transferase from murine fatty acid synthase is a promiscuous engineering tool for editing polyketide scaffolds" employed an acyltransferase (AT) domain swapping method that was previously developed in their laboratory (Ritter et al., Nat Chem 2022 = PMID: 35879443) and applied the strategy in a series of PKS systems (the venemyin PKS, a part of the pikromycin PKS, and a part of the erythromycin PKS) to alter the

PKS substrates. The authors constructed >30 expression plasmids and purified the corresponding proteins very carefully (e.g. immobilized metal affinity chromatography and ion exchange column chromatography). These engineered PKSs are mixed with various starter and extender substrates in vitro and polyketide productions were analyzed by LC-MS. However, for mass spectrometry data, it appears that all the data presented in the manuscript do not satisfy Communications Chemistry submission guidelines (please see below).

<https://www.nature.com/commschem/submit/submission-guidelines>:

Authors should also provide mass spectral data to support molecular weight identity. High-resolution mass spectral (HRMS) data are preferred, and when $m/z < 1,000$, the calculated and found values should be within 0.003. When $m/z > 1000$, we expect an experimental value within 1 ppm of the calculated value.

I also noticed the following comments in a few figure legends, which is not acceptable in general. Please use the same experimental conditions to create a figure for publication. Peak position is important to check chemical identity.

Figure S10 and Figure S12:

The peak shifting was caused by running the instrument at a different flow rate.

I also suggest that the authors purchase authentic standards and run them together in LC-MS analysis. For example, product 2 (= 4-Hydroxy-6-methyl-2-pyrone) is commercially available at reasonable price (<https://www.tcichemicals.com/JP/en/p/H0715>).

I am happy to review the manuscript again once the authors collected a solid set of experimental data, which is required to check if their experimental data support their conclusion.

Reviewer #3 (Remarks to the Author):

Grininger et al. report a study on engineering of polyketide synthase (PKS) assembly lines using an acyltransferase (AT) domain swapping strategy that the authors developed previously. This strategy depends primarily on the utilization of a promiscuous malonyl/acetyltransferase (MAT) domain of a murine fatty acid synthase (FAS), which appears to be a highly versatile tool for variable building block incorporation and is relatively compatible with other catalytic units in a hybrid assembly line.

To evaluate its application potential, the authors tested three different type I PKS systems that differ in size and are involved in the biosynthesis of the polyketides venemycin, pikomycin and erythromycin and accordingly produced a variety of polyketide analogs bearing substitution changes of a dedicated scaffold that arose from domain swapping with the promising FAS MAT.

The manuscript basically is a follow-up study, and the idea/strategy used in this study is not new and was well demonstrated by the authors in the previous works (e.g., Rittner, A.; et al. Nat. Chem. 2022, 14, 1000-1006). Technically it is good, as the methodology was tested and expanded to different PKS assembly lines and/or different positions in a previous studied system. If this manuscript is considered by the journal, the following issues need to be addressed for quality improvement.

Abstract. The authors claim that “Our study showcases MAT-based reprogramming of polyketide biosynthesis as a facile option for the regioselective editing of polyketide scaffolds”. In general, scaffold editing could be considered a bigger change such as size or folding pattern. The engineering works made in this study basically are based on the changes in side chain substitution rather than scaffold. Same as the below descriptions in the main text.

The authors could consider a shorter/concise introduction to PKS and AT engineering, as this is research article rather than a review.

Figure 3 is duplicated in the main text.

Reviewers' comments:

Reviewer #1:

Gringer and coworkers report the implementation of a promiscuous MAT domain to replace the native AT machinery of the venemycin PKS machinery and the native AT machinery of both Pik and DEBS platforms. Beginning with a small PKS system, MAT is swapped into the VEMS assembly line with adjusted linker regions to observe non-native starter and extender unit integration for an array of diverse pyrone products. MAT swaps are then tested in terminal modules of Pik and DEBS assembly lines where integration of MalCoA and FMMalCoA are observed. Their experiments show that MAT swaps result in promiscuous integration of starter and extender units, provided adequate care is taken to engineer the linking regions between PKS domains. The results in this manuscript are carefully prepared, and the arguments are thoughtfully laid out. this manuscript is therefore recommended for publication following the address of the following minor suggestions and concerns.

Reply: We appreciate the positive feedback and address the valuable points raised in your review in detail in the following.

Comment 1:

Line 61: On what basis do the authors claim that fluorination is the most important application of trans-AT machinery? There is no guarantee that structural similarity to solithromycin will result in medicinally useful compounds, especially since solithromycin was disqualified from clinical trials and nafithromycin has no fluorine. Please consider elaborating on this importance.

Reply: In this paragraph, we have given an overview on different AT domain engineering methods published in literature that includes a paper from our lab from 2022 (ref. 25). For the paper, we performed an AT/MAT swap and used fluorinated MalCoA analogs for chemoenzymatic derivatization of macrolactones. In the current manuscript, which is based on the work published in 2022, we are referring to the fluorination as the most important application of the AT/MAT swap we performed so far. This statement was not meant as a general statement, in the light that fluorination is the top derivatization strategy of polyketides. Regarding this, we have rephrased the statement slightly to avoid any misunderstanding. Precisely, we have changed the wording from "most importantly" to "such as".

The use of fluorinated MalCoA analogs in FAS/PKS hybrid enzymatic synthesis was inspired by the semi-synthetic drugs solithromycin and fluorithromycin. We think the specific reference to solithromycin is still useful, in spite of solithromycin's disqualification from therapeutic applications. Scientists may initiate follow-up studies on producing solithromycin-related analogs, where our work will potentially be of help.

Comment 2:

Line 78: A brief callout description of known substrates would be useful for the reader.

Reply: We agree with your suggestion of listing known substrates of the MAT domain to help the reader understand the idea of exploiting the enzymatic characteristics of MAT domain in PKS engineering.

We have adjusted the section from: "allowing a variety of substrates to be efficiently loaded" to "The MAT domain from mFAS is well characterized as a highly substrate promiscuous and kinetically fast enzymatic domain, enabling efficient loading of various substrates including non-, mono-, and disubstituted α -carboxyacyl-CoA extender substrates as well as acyl-CoA starter substrates with varying chain lengths and different stereochemistry and oxidation patterns at the α - and β -position (ref. 33, ref. 34)."

Comment 3:

Line 133: How can H1M0 have been purified and yet "not be obtained in purity"?

Reply: We changed "purified to "attempted to be purified" to avoid any misunderstanding and edited this section further.

Comment 4:

Line 138: It was assumed that the MAT could load the substrate onto M0 ACP despite the non-cognate interaction. This might be a reach. Is the lack of activity with AcCoA observed by H1M0 due to the MAT-ACPO disruption or the ACP0-KS1 disruption?

Please consider elaborating on why this assumption is valid.

Reply: Thank you for your comment on our assembly line constructs utilizing H1M0 and H2M0. H1M0 is a construct consisting of MAT and ACP from mFAS. These domains natively interact in mFAS, where the MAT catalyzes the transfer of an acetyl moiety from AcCoA (as well as the malonyl moiety of MalCoA) onto the ACP during a transacylation reaction. For H1M0 containing assembly lines, the non-cognate domain-domain interaction occurs during the chain translocation reaction between mFAS ACP and the downstream VEMS KS1. H2M0 is built from mFAS MAT and VEMS ACP0, such that, in H2M0 containing assembly lines, the non-cognate interaction occurs during transacylation.

We interpret the lack of activity in H1M0 containing PKS assembly lines as a result of the translocation reaction across the non-cognate interface. The translocation has been shown before by others and us to be rate-limiting, such that any interference in the interaction between ACP and downstream KS can hinder substrate turnover. In contrast to H1M0, in our study, H2M0 containing assembly lines were active, suggesting that the transacylation is less sensitive to the non-cognate interaction. This result is in line with previous studies that have shown that mFAS MAT can acylate a PKS-derived ACP domain (DEBS M6) (ref. 25).

We included a statement to the activities of H1M0 and H2M0 at the end of the paragraph, and included new references from the Abe lab, the Keatinge-Clay lab and my lab (ref.39, ref.43, ref. 46).

Comment 5:

Line 182: Sorry if I missed this, but consider explaining where 6H1, 8H2 and 10H3 constructs came from.

Reply: We are grateful for this comment, as it tells us that this sentence was misleading. We classify our constructs in groups H1, H2, and H3, which refer to domain borders for AT/MAT swapping. In sum, we have created 6 constructs of H1, 8 of H2, and 10 of H3. For information, the counting is not continuing, which is a result of how we proceeded in this project. We initially started with 12 constructs for each group. If we encountered difficulties in cloning constructs, we evaluated their need. In several cases, we came to the conclusion that a specific construct does not need to be pursued further without limiting the overall validity of our data. We believe that the selected constructs are sufficient to support the conclusions drawn in our study.

Nevertheless, we provide an overview of the initially designed constructs, the cloned expression plasmids, and the constructs that were finally tested for expression in *E. coli* in the supplementary information (Figure S5 and Table S4). This information was not considered as important enough to include it into the main text but is stated in the supporting information in the caption of Figure S5 "For module 2, a total of 36 constructs were designed, 12 constructs for each hybrid group (Table S4). Out of these, 31 constructs were cloned, and finally, 24 constructs were subjected to test expression in *Escherichia coli*."

Comment 6:

Line 194: Consider explaining your cutoff for sufficient protein to purify/further analysis.

Reply: We considered the expression yield as "insufficient" when it was <0.1 mg/L *E. coli* culture. In this case, cultivating >10 L culture or screening for optimized expression conditions would be required to get a sufficient amount of protein for our quality control and analysis protocol described in the method section. We added this information to the main text: "Sodium dodecyl-sulfate polyacrylamide gel electrophoresis (SDS-PAGE) analysis revealed that just certain constructs express, albeit at insufficient quantities (<0.1 mg/L *E. coli* culture) to allow protein isolation and analysis (Figures S7 and S8)."

Comment 7:

Figure 3B: The LCMS plots indicate that compounds 2, 3, and 6 were successfully produced, but the checks/ticks indicate the production of 2, 3, and 5. Likewise, the commentary text (page 5, line 150 and page 18, line 587) indicates 2, 3, and 4. Please clarify. Maintain the same order in the legends as in the LCMS chromatogram traces. Propionyl and 4-hydroxybutyryl-CoA are swapped in order.

Reply: Many thanks for careful reading. This misleading information resulted from accidentally uploading Figure 3 twice (old and new versions). The compounds and traces are now in the correct order in the figures, legends, as well as in the main text.

Comment 8:

Line 205: Consider adding a brief comment on the relative production of 5-methylvenemycin compared to venemycin production.

Reply: In this manuscript, LC-MS analysis was performed to verify the production of venemycin derivatives but not for quantification. The peak intensities for venemycin and the methylated derivative may not be comparable and could be influenced e.g. by the solubility of the compounds during the extraction process. That is why we want to refrain from making statements about relative production. Quantification could be achieved by running calibration curves, but both substances are not commercially available and chemical synthesis would be too elaborate given our aim to evaluate the applicability of AT/MAT-swaps for enabling loading of non-native extender substrates.

Comment 9:

Line 209: Perhaps not relevant for publication, but I'm curious if you tested a MAT swap in AT1 and AT2 simultaneously to see if you could produce 8?

Reply: Since we were not able to obtain a functional MAT-swapped module 2 albeit testing 24 constructs for expression, we, regrettably, cannot provide experimental data.

Comment 10:

Line 228: This "loading and extension" activity by M1 seems reminiscent of module stuttering, as mentioned by Keatinge-Clay 2012 and others. Consider commenting on this.*

Reply: Indeed, our proposed mechanism and module stuttering have some aspects in common. Module stuttering is characterized by two subsequent elongation reactions catalyzed by the same module and an intramodular intermediate transfer from the ACP to its upstream KS between the two elongation steps. In our case, we propose intramodular intermediate transfer followed by a single elongation reaction within one module. That is why we would refrain from calling it module stuttering, but see the similarity in the intramodular chain transfer reaction. To comment on this similarity, we added the following sentence to the main text and added a reference to Keatinge-Clay 2017 (ref. 52): "We propose that the propionyl loaded ACP transfers propionyl back to the intramodular KS domain, instead of translocating it to the downstream module, which is a mechanism observed in some modular PKSs in the context of "module stuttering" (ref 52)."

Comment 11:

Line 258: Consider elaborating a little more on the relevance of fluoromethylmalonyl-CoA.

Reply: The use of fluoromethylmalonyl-CoA (FMMalCoA) substrate was first described by our lab in the context of a mono-modular FAS/PKS hybrid system derived from DEBS (ref. 25). We used FMMalCoA to improve product stability because we observed degradation of fluorinated macrolactones received with the substrate fluoromalonyl-CoA (FMalCoA). In addition, the use of FMMalCoA made motifs available as found in solithromycin. As we argue above, the chemoenzymatic incorporation of the methyl-fluoro unit may be valuable for producing analogs of solithromycin (see our reply to your comment 1) or fluorithromycin that also comprises a F/Me-substitution within the core scaffold. We further consider it interesting that with FMMalCoA we could show the direct disubstitution of macrolactone scaffolds. In nature, dimethylation is often found as a structural motif in polyketides and (for cis-AT PKSs) is usually accomplished by methyltransferase (MT) domains after elongation with mono-substituted substrates (i.e. MMalCoA). It has been suggested by the Keasling lab that naturally dimethylation also occurs directly from dimethylmalonyl-CoA substrates in the biosynthesis of some polyketides (S. Poust et al., 10.1002/anie.201410124). However, direct dimethylation has not been shown in an engineering approach. Thus, our results highlighted direct disubstitution as engineering approach, and, in addition, gave interesting mechanistic insight, e.g., showing that decarboxylation must precede carbon-carbon-bond formation.

We have added more information on FMMalCoA to the subchapter "AT/MAT-swaps in mono-modular PIKS system" as follows: "The use of di-substituted FMMalCoA was first described in the context of a FAS/PKS hybrid of DEBS M6 (ref. 25). FMMalCoA gives access to motifs found in solithromycin. Additionally, the incorporation of the F/Me-

substitution demonstrates the ability of direct disubstitution of macrolactone scaffolds.”

Moreover, we have included a more detailed discussion on the chances of MAT for the derivatization of polyketides in the conclusion section: “It should also be noted that this study demonstrates the ability of three modules of modular PKSs, PIKS M5, and DEBS M5 and M6 (also previously shown (ref. 25)) to elongate a growing polyketide chain with fluoromethyl-malonyl. This suggests that by harnessing the MAT domain, the direct incorporation of disubstitutions into the polyketide scaffold appears to be a viable engineering strategy. Since most natural compounds are not directly administered as medicines, but derivatized for use in therapy (ref. 58), derivatization is important to optimize a natural product’s binding to its biological target, pharmacokinetic properties, and bioavailability. Based on our data, AT/MAT-swaps could constitute a viable engineering strategy for enabling not just mono-substitutions (ref. 59 and ref. 60), but programmable and site-selective disubstitutions, such as fluoromethylations (ref. 61) shown in this study, or as a perspective gem-dimethylations (ref. 62).”

Comment 12:

Line 264: This is unexpected indeed. Consider describing the implications of this somewhere in the manuscript. For example, does this suggest that the chimeragenesis approach might not always be necessary? Or that the Pik M5 AT could serve as an alternative to MAT?

Reply: We thank the reviewer for pointing out the unexpected results of the formation of 2-fluoro-10-dml (**17**) from incubation of PIKS activated pentaketide and FMMA_lCoA with PIKS M5-TE. For PIKS M5 AT, MMa_lCoA is the native substrate resulting in the formation of 10-dml (**14**). Intriguingly, while Ma_lCoA is not processed as a substrate, PIKS M5 AT is “leaky” for the di-substituted FMMA_lCoA. Indeed, data indicates substrate tolerance that may be exploited for polyketide derivatization using a native PKS system.

In a previous study, we determined the kinetic parameters for MAT-mediated transacylation of fluoromethylmalonyl onto mFAS ACP with k_{cat} 12.1 s⁻¹, K_S 2.7 μM, K_{ACP} 24.6 μM and k_{cat}/K_S 4.5 x 10⁶ M⁻¹s⁻¹ and compared them with data for DEBS M6 AT-mediated transacylation of the natural substrate MMa_lCoA to DEBS M6 ACP (0.29, 50.8, 210, 5.8 x 10³) (ref. 25). In another study, we have determined parameters for PIKS M5 AT-mediated transacylation of the natural substrate MMa_lCoA to PIKS M5 ACP (0.75, 96.2, 330, 7.8 x 10⁴) (ref. 32). Based on these data, the enzymatic efficiency of mFAS MAT for the non-canonical substrate FMMA_lCoA is 775 times higher than that of DEBS M6 AT and 58 times higher than that of PIKS M5 AT for their natural substrates MMa_lCoA. We have collected more data showing the high efficiency of mFAS MAT for other non-canonical substrates (ref. 33). From this comparison, we do not assume that AT domains of PKSs offer a broadly applicable solution for installing fluoromethyl or other modifications in polyketides. Rather, the good performance of PIKS M5, compared to hybrid module, may not be due to the AT per se, but rather to other factors such as the integrity of the native protein or the cognate interaction between PIKS M5 AT and PIKS M5 ACP. A kinetic analysis of PIKS M5 AT would certainly provide more insights here. Once again, we would like to thank you for discussing this point.

We have included an additional paragraph to the conclusion chapter to address this point appropriately.

Comment 13:

Line 283: Consider explaining why swapping M5 would cause such a drop in purified protein.

Reply: For VEMS PKS system, a comprehensive domain boundaries study for AT/MAT-swaps was conducted with the aim to establish engineering guidelines for the generation of hybrid systems. Many hybrid designs did not lead to protein at all or the protein was inactive, however, we also found that a domain boundaries design (termed H1), previously described by our lab for the DEBB M6 hybrids, shows the highest success rates. We therefore decided to apply the H1 AT/MAT-swap design for DEBS M5, leading to construct DEBS M5*-M6-TE, which unexpectedly led to just low protein amounts. We can just speculate on the origin of this difference, but the bottom line is that the AT/MAT swap is invasive, and despite the guidance this study can give, each hybrid design will remain an individual problem and subject to its specific optimizations. We would like to note that highlighting such difficulties is a strength of in vitro studies. As we mention in the conclusion section, trans-AT engineering strategies, using MAT

as a separate protein, may be powerful in this respect and solve protein production problems.

We have included a statement for the low protein yield of hybrid M5*-M6-TE in the conclusion chapter: "Intriguingly, even similar hybrid designs can lead to different outcomes, as demonstrated by the varying recombinant production levels of DEBS M5*-M6-TE (low protein yield) and DEBS M5-M6*-TE (high protein yield) (Figure S17). We consider it a strength of in vitro studies to highlight such pitfalls, in this case, to illustrate that each protein design, regardless of how similar it is, will be subject to its own specific requirements."

In addition, we have rephrased the final remark of the conclusion chapter: "Limitations may arise from the intertwined KS-LD-AT fold, and the general intricate structure of the PKS module."

Comment 14:

Line 323: Shifts in retention time can not be conclusive evidence of regioselective differences. While differences certainly suggest that regioselectivity has changed, only NMR and X-Ray experiments conclusively prove this. Please use language that "suggests" the production of the supposed analogs. Different retention times are "consistent with" regioisomers.

Reply: We absolutely agree with the reviewer's comments that comprehensive analytical elucidation (e.g. NMR) is required to confirm the formation of C-2 and C-4 regioisomers of 3-hydroxy-demethyl-narbolide **19/21** and fluoro-3-hydroxy-narbolide **20/22**, respectively. And we regret that the wording we used was not appropriate in this regard.

The study has aimed to exploit the scope of FAS/PKS hybrids to different PKS assembly lines and more complex systems. As such it is focused mainly on the proteins, while we consider the production of compounds as read-out for the engineering success. We have toned down the statement on the chemical identity of the compounds. When presenting and discussing data received from DEBS M5*-M6-TE and DEBS M5-M6*-TE, we now carefully refer to that chemical structures of compounds are derived from HPLC-HRMS, and NMR analysis would be required for confirmation.

We have included the following section in the manuscript: "We note that the chemical identity of compounds was determined by HPLC-HRMS, which does not give conclusive evidence of regioselectivity of polyketide modification. Given this limitation, NMR analysis of compounds would be necessary to confirm the modification at position 4 without ambiguity."

Additionally, we have adapted the language to emphasize that further structure elucidation is required. As such, we have changed "production of C-2 analogs" to "we produced different regioisomers."; and included another statement to point out required NMR analysis: "However, again we note the NMR analysis will need to confirm suggested compound structures."

Comment 15:

Line 353: "led" instead of "let"

Reply: We thank the reviewer for carefully reading the manuscript. "let" was changed to "led".

Comment 16:

Line 369: "Broad flexibility of the MAT domain". Ok, but this needs context. It is only broader with respect to fluorinated malonyl-CoAs. There are likely many extenders MAT cannot use that native or engineered ATs can. How well would this approach work with assembly lines that contain malonyl-CoA specific ATs that might compete with the FMM-CoA? Or indeed assembly lines that are as promiscuous as the Pik Module 5 AT?

Reply: Again, we are grateful for this comment. Especially, in the context of the unexpected acceptance of FMMalCoA by PIKS M5 AT, the value of MAT seems questionable. As we have already discussed above in our reply to your comment 12, the mFAS MAT has unmatched high efficiency in transacylating non-canonical substrates with high efficiency. To quantify this property, we analyzed ten non-canonical substrates for the MAT (ref. 33). The enzymatic efficiency of all the various substrates tested in this study exceeds the efficiencies of PKS ATs (ref. 32) by several folds

showing that the MAT has superior properties, which led us conclude the term “broad substrate flexibility of the MAT domain” to describe its polyspecificity.

From the set of acyl-CoA esters analyzed in kinetic properties, four were extender substrates; MalCoA, MMalCoA, FMalCoA and FMMalCoA. While we do not have kinetic data for other extender substrates that were described in the work of others (e.g. the Williams lab (E. Kalkreuter et al., doi: 10.1021/jacs.8b10521) or the Erb lab (K. Geyer et al., doi: 10.1002/cbic.202000112)), we speculate that also those would be accepted with reasonable efficiency based on the relatively large space of malonyl in the binding pocket of mFAS MAT (pdb: 5my0) and the observed plasticity of the MAT binding pocket (ref. 33).

We propose that assembly lines that contain MalCoA-specific AT domains (e.g. PKS M2) would not incorporate FMMalCoA at the respective position within the polyketide scaffold. FMMalCoA is similar in size to MMalCoA, which is naturally excluded from processing by MalCoA-specific ATs. Nonetheless, the broad flexibility of the MAT domain will result in the production of a compound mixture (see bi-modular DEBS3 system) when multiple substrates are present. Ultimately, compound purification and isolation after enzymatic synthesis via a FAS/PKS hybrid system is required.

Since we have already added a paragraph about the kinetic properties of the MAT domain in the conclusion section (see our reply to your comment 12) giving more background on the broad substrate flexibility of the MAT domain, we believe that further context to this statement is not necessary.

Comment 17:

Line 378: If MAT needs to be engineered to be more specific within a biosynthetic pathway, how is it superior to assembly line ATs, which are already extender unit specific?

Reply: We appreciate the reviewer’s note on how mFAS MAT domain is superior to substrate-specific assembly line ATs. As mentioned above, our lab has extensively studied the MAT domain. We have come to the conclusion that the MAT domain is a promising domain for PKS engineering as it combines several key features. These include the structural integrity of the fold, fast transacylation kinetics that are up to three orders of magnitude higher than those of native AT domains, heterologous expression in *E. coli*, and foremost its substrate polyspecificity. This not only enables the use of various extender substrates but also of different starter substrates to prime the enzymatic synthesis at different positions (see Fig. 6, VEMS M1*).

We have stated in our conclusion that in complex PKS assembly lines, the broad flexibility of the MAT domain will require separation of the product mixture. In order to reduce the need for product separation, we proposed the possibility to further develop the AT/MAT-swap approach by engineering specialized MAT domains that prefer a certain substrate while retaining MAT domain’s advantageous characteristics (fold stability and fast transacylation kinetics). With its properties of being fast and substrate promiscuous, the MAT domain features properties of evolutionary ancestor proteins, which are supposed to be good leads for starting engineering approaches (ref. 63) (a phylogenetic analysis of the MAT has been performed by us previously, ref. 25). Not least, the approach of ancestral sequence reconstruction is built on the idea to predict sequences of ancient proteins to reveal a more stable and functionally diverse variant that can then be harnessed for protein engineering purposes.

The ability to engineer the MAT domain was previously demonstrated by our lab (ref. 25). Thus, we claim that the use of a fast and promiscuous domain that is accessible for further engineering toward specialized mutants is superior to an approach that utilizes slow and substrate-specific assembly line ATs and tries to alter substrate-specificity.

We have added the following statement to the conclusion section: “We would like to emphasize that broad substrate acceptance and high stability, both features observed for mFAS MAT, are generally regarded as good starting points for the engineering of proteins towards new functions.”

Comment 18:

Supplemental Figure 1B: why does the stereochemistry of the alpha carbon change after elongation and again after reduction? Shouldn’t this center remain unchanged?

Reply: The stereochemistry of the alpha carbon can change during the KS-catalyzed Claisen condensation reaction (from 2S to 2R). The KR domain of DEBS module 1 belongs to the B2 type KR domains which catalyze epimerization of the C-2 methyl

group (from 2R to 2S) followed by reduction to give the (2S,3R)-diketide. The mechanism of stereocontrol in DEBS module 1 is reviewed in K. Weissmann, <https://doi.org/10.3762/bjoc.13.39>.

Comment 19:

Supplemental Figure 3: The connection of substrates to the KS should occur on a straight line, as this bond is the direct carbon-sulfur bond of cysteine rather than a pantetheinylated sulfur bond of an ACP.

Reply: We agree with the reviewer and now use straight lines for substrate connection to the KS domains in supplemental figures 1 and 3.

Comment 20:

Supplemental Figure 14: it is optimistically described that the presence of 10 in a sample expecting to produce 4 “cannot be ruled out”, but given that other experiments (Figure S14C, right) present a peak with an identical retention time, it seems extremely likely that this peak is in fact 10. Coupled with the fact that this second peak could not be produced in the absence of mmCoA, this seems obvious. It would not be a loss to say so.

Reply: We are grateful for the thorough examination and agree with the interpretation of the data. We updated the figure (now Figure S13) using LC-HRMS data. We now show the main products for each reaction mixture and show all possible side products. We are more precisely referring to the possible side product formation, but prefer to remain cautious in our statements, given that just NMR analysis can finally unambiguously confirm chemical identity.

Reviewer #2:

The authors of this manuscript titled “The malonyl/acetyl-transferase from murine fatty acid synthase is a promiscuous engineering tool for editing polyketide scaffolds” employed an acyltransferase (AT) domain swapping method that was previously developed in their laboratory (Ritter et al., Nat Chem 2022 = PMID: 35879443) and applied the strategy in a series of PKS systems (the venemyin PKS, a part of the pikromycin PKS, and a part of the erythromycin PKS) to alter the PKS substrates. The authors constructed >30 expression plasmids and purified the corresponding proteins very carefully (e.g. immobilized metal affinity chromatography and ion exchange column chromatography). These engineered PKSs are mixed with various starter and extender substrates *in vitro* and polyketide productions were analyzed by LC-MS. However, for mass spectrometry data, it appears that all the data presented in the manuscript do not satisfy Communications Chemistry submission guidelines (please see below).

<https://www.nature.com/commschem/submit/submission-guidelines>:

Authors should also provide mass spectral data to support molecular weight identity. High-resolution mass spectral (HRMS) data are preferred, and when $m/z < 1,000$, the calculated and found values should be within 0.003. When $m/z > 1000$, we expect an experimental value within 1 ppm of the calculated value.

I am happy to review the manuscript again once the authors collected a solid set of experimental data, which is required to check if their experimental data support their conclusion.

Reply: We are grateful for the feedback and pointing out the importance of providing high-resolution mass spectral data in line with *Nature Communications Chemistry* submission guidelines. We have taken this as an opportunity to repeat the chemoenzymatic synthesis of all compounds and analyzed them by LC-HRMS. The extracted ion chromatograms were obtained by searching for the calculated weight of the expected molecular formula with an error of 0.003. Accordingly, the following figures in the main text and the supporting information have been updated: Figure 3, Figure 5, Figure 6, Figure 7, and Figure 9; Figure S10, Figure S12, and Figure S13. Additionally, two new figures showing mass spectra data have been appended to the supporting information: Figure S16 and Figure S20.

Comment 1:

I also noticed the following comments in a few figure legends, which is not acceptable in general. Please use the same experimental conditions to create a figure for publication. Peak position is important to check chemical identity.

Figure S10 and Figure S12: The peak shifting was caused by running the instrument at a different flow rate.

Reply: Figure S12 was deleted during revision because we consider this additional information as not relevant. Accordingly, we have removed the comment about peak shifting from the legend of Figure S10. It does no longer apply to the new data obtained by repeating the experiments for generating high-resolution mass spectral data.

Comment 2:

I also suggest that the authors purchase authentic standards and run them together in LC-MS analysis. For example, product 2 (= 4-Hydroxy-6-methyl-2-pyrone) is commercially available at reasonable price (<https://www.tcichemicals.com/JP/en/p/H0715>).

Reply: We utilized commercial 4-hydroxy-6-methyl-2-pyrone (product 2) to optimize extraction and LC-MS analysis conditions for both product 2 and its derivatives. To further enhance the clarity of our findings, we have included LC-MS analysis data for commercial 4-hydroxy-6-methyl-2-pyrone in the supporting information (Figure 13B) for comparison with product 2 generated by the engineered PKSs.

Our study is focused on the applicability of the AT/MAT swap. Throughout the study, we focus on the proteins (yield in recombinant expression, stability, activity), and we utilize LC-MS analysis as a mean to monitor the functionality of the hybrid proteins. It was not our focus to quantify the produced compounds, such that we did not use product 2 as an internal standard in our reaction mixtures.

Reviewer #3:

Gringer et al. report a study on engineering of polyketide synthase (PKS) assembly lines using an acyltransferase (AT) domain swapping strategy that the authors developed previously. This strategy depends primarily on the utilization of a promiscuous malonyl/acetyltransferase (MAT) domain of a murine fatty acid synthase (FAS), which appears to be a highly versatile tool for variable building block incorporation and is relatively compatible with other catalytic units in a hybrid assembly line. To evaluate its application potential, the authors tested three different type I PKS systems that differ in size and are involved in the biosynthesis of the polyketides venemycin, pikomycin and erythromycin and accordingly produced a variety of polyketide analogs bearing substitution changes of a dedicated scaffold that arose from domain swapping with the promising FAS MAT.

The manuscript basically is a follow-up study, and the idea/strategy used in this study is not new and was well demonstrated by the authors in the previous works (e.g., Rittner, A.; et al. Nat. Chem. 2022, 14, 1000-1006). Technically it is good, as the methodology was tested and expanded to different PKS assembly lines and/or different positions in a previous studied system. If this manuscript is considered by the journal, the following issues need to be addressed for quality improvement.

Reply: Thank you for your thoughtful review of our manuscript. We agree that this manuscript builds upon our previous work (Rittner, A.; et al. Nat. Chem. 2022, 14, 1000-1006, ref. 25), where we first demonstrated a MAT swap in the context of PKS engineering. In this current manuscript, we have further developed this concept by exploring its broader utilization in PKS engineering. This includes investigating its application in different PKSs, exploring the potential for multiple swaps per polypeptide, studying non-terminal and loading modules, utilizing the loading capability of MAT, and proposing universal guidelines for its implementation (e.g., considering H1 as not only the best for DEBS M6 but as a universal guideline). These aspects are crucial for assessing the feasibility of the MAT swap in PKS engineering, even though the overall concept may not be entirely new. We are grateful for your recommendations for improving the manuscript and have addressed the raised issues in a detailed point-by-point reply.

Comment 1:

Abstract. The authors claim that “Our study showcases MAT-based reprogramming of polyketide biosynthesis as a facile option for the regioselective editing of polyketide scaffolds”. In general, scaffold editing could be considered a bigger change such as size or folding pattern. The engineering works made in this study basically are based on the changes in side chain substitution rather than scaffold. Same as the below descriptions in the main text.

Reply: Thank you for raising this point. We agree that “scaffold editing” can be misleading, and that it is important to specify the level of modification we can introduce to the polyketide scaffold in the abstract of this manuscript. We now use a clearer wording to avoid misunderstandings. Instead of “scaffold editing”, we now term it “editing of residues decorating the polyketide scaffold”, which makes clear that our engineering is restricted to the residues that decorate the scaffold rather than altering the size of the polyketide scaffold.

Comment 2:

The authors could consider a shorter/concise introduction to PKS and AT engineering, as this is research article rather than a review.

Reply: We have taken your suggestion into consideration and have shortened the introduction by nearly 150 words (15%). Specifically, we have trimmed down the section in which AT engineering approaches are highlighted by examples from the literature.

Comment 3:

Figure 3 is duplicated in the main text.

Reply: We thank the reviewer for carefully checking the manuscript and deleted the duplication of Figure 3.

REVIEWERS' COMMENTS:

Reviewer #1 (Remarks to the Author):

The revised manuscript addresses most of this reviewer's comments and suggestions from the original submission. Technically, I am not sure the presentation of macrolactone structural identity is appropriate when the regioselectivity of extender unit incorporation is unknown. Although I appreciate the author's disclaimer to this point, I would prefer the structures presented in the schemes to reflect the ambiguity, but I will defer to the editor's judgment on that issue.

Reviewer #2 (Remarks to the Author):

Communications Chemistry (Submission ID COMMSCHEM-24-0049)

The malonyl/acetyl-transferase from murine fatty acid synthase is a promiscuous engineering tool for editing polyketide scaffolds

Buyachuihan L et al

The authors of this manuscript titled “The malonyl/acetyl-transferase from murine fatty acid synthase is a promiscuous engineering tool for editing polyketide scaffolds” employed an acyltransferase (AT) domain swapping method that was previously developed in their laboratory (Ritter et al., Nat Chem 2022 = PMID: 35879443) and applied the strategy in a series of PKS systems (the venemyin PKS, a part of the pikromycin PKS, and a part of the erythromycin PKS) to alter the PKS substrates. I recommend to publish this manuscript after the revision suggested below.

1) Please show both the calculated and found values in all related figures and calculate mass errors (ppm) as indicated in the author guideline (see below). The current manuscript contains the calculated values only.

<https://www.nature.com/commschem/submit/submission-guidelines>:

Authors should also provide mass spectral data to support molecular weight identity. High-resolution mass spectral (HRMS) data are preferred, and when $m/z < 1,000$, the calculated and found values should be within 0.003. When $m/z > 1000$, we expect an experimental value within 1 ppm of the calculated value.

Reviewer #3 (Remarks to the Author):

The authors have now satisfactorily addressed the concerns from the reviewers. a minor issue: given the fact that the word "residues" is often referred as to the units of a peptide natural product, the reviewer would suggest the change of the description to "editing of carboxylic building blocks to decorate the polyketide scaffold".

Editors' comments:

Comment 1:

We therefore invite you to revise your paper one last time to address the remaining concerns of our reviewers, particularly, add a comment to reflect the structural ambiguity in the Figure 9 legend.

Reply: Thank you for encouraging us to highlight in Figure 9 legend that the identification of compounds was based on HPLC-HRMS data. We have included the following explanation to the Figure 9 legend:

“We note that the chemical identity of compounds **19**, **20**, **21** and **22** was determined by HPLC-HRMS, which does not give conclusive evidence about regioselectivity of polyketide modification.”

Comment 2:

... add mass errors in the relevant SI figures.

Reply: In accordance with *Nature Communications Chemistry* author guidelines regarding mass spectral data and to demonstrate the successful formation of compounds, we have added mass errors (in ppm) to all high-resolution mass spectra shown in the supplementary information. For this, we have updated the figures legends of figures S10, S12, S13, S16 and S20. In addition, we have included the following statement to the Methods section of the manuscript to highlight the generation of the EIC blots: “For EIC blot display, the m/z tolerance was set to 0.003.”

Reviewers' comments:

Reviewer #1:

The revised manuscript addresses most of this reviewer's comments and suggestions from the original submission. Technically, I am not sure the presentation of macrolactone structural identity is appropriate when the regioselectivity of extender unit incorporation is unknown. Although I appreciate the author's disclaimer to this point, I would prefer the structures presented in the schemes to reflect the ambiguity, but I will defer to the editor's judgment on that issue.

Reply: We thank the reviewer for reading our manuscript once again and for appreciating our efforts to improve the original version of our draft. To address your concern regarding the presentation of macrolactone structures, we have added a comment in the Figure 9 legend to emphasize that structural identity of regioisomers remains ambiguous without further structural elucidation.

Reviewer #2:

The authors of this manuscript titled “The malonyl/acetyl-transferase from murine fatty acid synthase is a promiscuous engineering tool for editing polyketide scaffolds” employed an acyltransferase (AT) domain swapping method that was previously developed in their laboratory (Ritter et al., Nat Chem 2022 = PMID: 35879443) and applied the strategy in a series of PKS systems (the venemyin PKS, a part of the pikromycin PKS, and a part of the erythromycin PKS) to alter the PKS substrates. I recommend to publish this manuscript after the revision suggested below.

Reply: We appreciate the positive feedback on our revised manuscript and are pleased that our study has been proposed for publication in *Nature Communications Chemistry*. We have addressed the comment regarding compliance with the author guidelines for mass spectral data below.

Comment 1:

1) Please show both the calculated and found values in all related figures and calculate mass errors (ppm) as indicated in the author guideline (see below). The current manuscript contains the calculated values only.

Reply: We have carefully updated all related figures to comply with the author guidelines of *Nature Communications Chemistry*. Precisely, we have now clearly labelled calculated and found m/z values of all products in the respective manuscript figures and SI figures. Additionally, we have updated the figures legends of the SI figures that display HR-mass spectra by calculating the mass error (in ppm) for each found m/z value (e.g., deviation: -0.8 ppm). For this, we have revised manuscript figures: Figure 3 and Figure 5, and SI figures: Figure S10, Figure S12, Figure S13, Figure S16, Figure S18, Figure S19 and Figure S20 as well as their figure legends, respectively. As mentioned above, we have included a statement in the manuscript Method section to explain the settings chosen for EIC display.

Reviewer #3:

The authors have now satisfactorily addressed the concerns from the reviewers.

Reply: We are delighted with the positive response to our revised manuscript and thank the reviewer for acknowledging our efforts of refining the original version of our manuscript.

Comment 1:

a minor issue: given the fact that the word "residues" is often referred as to the units of a peptide natural product, the reviewer would suggest the change of the description to "editing of carboxylic building blocks to decorate the polyketide scaffold".

Reply: Thank you for raising this point. We agree that the term "residues" which is mostly used for describing peptides and proteins, can be confusing to the reader. However, we think, the suggested wording "editing of carboxylic building blocks to decorate the polyketide scaffold" would be unclear for the broader readership. This is why we revised the wording in the manuscript to "editing substituents decorating the polyketide scaffold".